# Promoting Ensemble Diversity with Interactive Bayesian Distributional Robustness for Fine-tuning Foundation Models

Ngoc-Quan Pham [* 1]  Tuan Truong [* 1]  Quyen Tran [1]  Tan Nguyen [1 2]  Dinh Phung [1 3]  Trung Le [3]

## Abstract

We introduce Interactive Bayesian Distributional Robustness (IBDR), a novel Bayesian inference framework that allows modeling the interactions between particles, thereby enhancing ensemble quality through increased particle diversity. IBDR is grounded in a generalized theoretical framework that connects the distributional population loss with the approximate posterior, motivating a practical dual optimization procedure that enforces distributional robustness while fostering particle diversity. We evaluate IBDR's performance against various baseline methods using the VTAB-1K benchmark and the common reasoning language task. The results consistently show that IBDR outperforms these baselines, underscoring its effectiveness in real-world applications.

## 1. Introduction

Tackling uncertainty remains one of the most challenging problems in deep learning. This uncertainty arises from the real world's inherent randomness and noisy, complex data. To address this challenge, Bayesian inference offers a powerful solution by providing a probabilistic framework that enables reasoning under uncertainty. A particularly practical approach within Bayesian inference involves particle sampling techniques, which are well-suited for scenarios requiring multiple models. Notable methods include Stochastic Gradient Langevin Dynamics (SGLD) (Welling & Teh, 2011a), Hamiltonian Monte Carlo (HMC) (Neal, 1996), Stochastic Gradient HMC (SGHMC) (Chen et al., 2014), and Stein Variational Gradient Descent (SVGD) (Liu & Wang, 2016). However, a key limitation of these methods is the computational and storage overhead associated with

maintaining multiple models, especially for large-scale architectures. To mitigate this issue, variational inference (VI) techniques have been developed to approximate the true posterior distribution with tractable approximations known as variational posteriors. These methods optimize a variational lower bound, significantly reducing computational costs while effectively capturing uncertainty. Key contributions in this domain include Kingma & Welling (2013); Kingma et al. (2015), and Blundell et al. (2015), who extended Gaussian variational posterior approximations to neural networks. Additionally, Gupta & Nagar (2018) introduced greater flexibility in posterior approximations, enhancing the applicability of VI techniques.

Traditional Bayesian inference faces a limitation: the independent sampling of model particles from the posterior often fails to capture interactions and can lead to particle collapse into a single mode. To overcome this, we introduce a novel framework, Interactive Bayesian Distributional Robustness (IBDR), which establishes a joint distribution over independent posteriors and integrates a divergence loss to model this interaction and encourage particle diversity. To further enhance robustness, we leverage Wasserstein-based distributional robustness optimization (Gao & Kleywegt, 2023; Blanchet et al., 2019; Sinha et al., 2018), as formalized in Theorem 4.1. This extension generalizes distributional robustness optimization (DRO) to accommodate more general risk functions and product joint distributions, ensuring a balance between robustness and diversity. The resulting framework improves ensemble performance by preventing mode collapse while maintaining computational efficiency. To demonstrate the effectiveness and versatility of our framework, we conduct experiments on the image classification task with ViT (Dosovitskiy et al., 2020) and the commonsense reasoning task on LLaMA-2 (Touvron et al., 2023), demonstrating significant improvements over baseline methods.

Our contributions are as follows: **(i)** We introduce a novel Bayesian framework that explicitly models **interactions between model particles during training**. Leveraging distributional robustness, we provide a theoretical analysis of this interactive framework, generalizing existing results to product distribution spaces. **(ii)** Building on this analysis,

---

[*]Equal contribution [1]Qualcomm AI Research, Qualcomm Vietnam Company Limited [2]National University of Singapore, Singapore [3]Monash University, Australia. Correspondence to: Trung Le <trunglm@monash.edu>.

*Proceedings of the 42$^{nd}$ International Conference on Machine Learning*, Vancouver, Canada. PMLR 267, 2025. Copyright 2025 by the author(s).

we propose a practical framework that ensures both **ensemble diversity** and **distributional robustness**. We validate our approach by fine-tuning ViT (Dosovitskiy et al., 2020) on the VTAB-1K classification benchmark and LLaMA-2 (Touvron et al., 2023) on a commonsense reasoning task, demonstrating significant performance improvements.

## 2. Related Works

### 2.1. Bayesian Neural Networks

**Variational Inference.** This approach estimates the posterior distribution by selecting a specific approximation and refining a variational lower bound. Graves (2011) introduced a Gaussian variational posterior for neural network weights, further developed in Kingma & Welling (2013); Kingma et al. (2015); Blundell et al. (2015) using the reparameterization trick for deep latent variable models. Extensions to improve posterior flexibility include Caterini et al. (2021), who used normalizing flows, and Louizos & Welling (2017) and Gupta & Nagar (2018), who employed a matrix-variate Gaussian. Other studies have explored structured variational Gaussian posteriors, such as Kronecker-factored approximations (Rossi et al., 2020; Eschenhagen et al., 2023), Gaussian score matching (Modi et al., 2023), and non-centered or rank-1 parameterizations (Ghosh et al., 2018; Dusenberry et al., 2020). Hybrid approaches combining Variational Inference and Markov Chain Monte Carlo (MCMC) have gained popularity, such as Alexos et al. (2022), which used latent variable averaging to speed up mixing. Applications of Variational Inference to modern architectures, like Vision Transformers (Zhang et al., 2021), highlight its role in uncertainty-aware deep learning.

**Markov Chain Monte Carlo (MCMC).** This approach allows sampling multiple models from the posterior distribution, commonly used for neural network inference via Hamiltonian Monte Carlo (HMC) (Neal, 1996). However, HMC requires full gradient estimation, which is computationally expensive. Stochastic Gradient HMC (SGHMC) (Chen et al., 2014) uses stochastic gradients for scalability and solution exploration. Alternatively, Stochastic Gradient Langevin Dynamics (SGLD) (Welling & Teh, 2011b) applies Langevin dynamics in the stochastic gradient setting. Stein Variational Gradient Descent (SVGD) (Liu & Wang, 2016) uses particles to approach the posterior distribution. MCMC methods, unlike variational methods, can be costly due to the need to store multiple models. Nonetheless, SGHMC, SGLD, and SVGD asymptotically sample from the posterior with infinitely small step sizes.

### 2.2. Flat Minimizers

Flat minimizers improve neural network generalization by helping models find broader local minima, making them more robust to training-test set differences (Jiang et al., 2020; Petzka et al., 2021; Nguyen et al., 2023a). The link between generalization and minimum width has been explored both theoretically and empirically (Hochreiter & Schmidhuber, 1994; Neyshabur et al., 2017; Dinh et al., 2017; Fort & Ganguli, 2019). Various methods for finding flat minima have been proposed (Pereyra et al., 2017; Izmailov et al., 2018; Foret et al., 2021; Nguyen et al., 2023a). Studies by Keskar et al. (2017), Jastrzebski et al. (2017), and Wei et al. (2020) examine how batch size, learning rate, gradient covariance, and dropout affect flatness. Some methods also use regularization terms in the loss function to promote wide local minima (Pereyra et al., 2017; Zhang et al., 2018; 2019), such as softmax entropy penalties (Pereyra et al., 2017) and distillation losses (Zhang et al., 2018; 2019).

Among the flat minimizers, Sharpness-Aware Minimization (SAM) (Foret et al., 2021) has gained attention for its effectiveness and scalability across tasks like domain generalization (Cha et al., 2021; Wang et al., 2023; Zhang et al., 2023), federated learning (Caldarola et al., 2022; Qu et al., 2022), Bayesian networks (Nguyen et al., 2023b; Möllenhoff & Khan, 2023), and meta-learning (Abbas et al., 2022). SAM has also improved generalization in both vision models (Chen et al., 2022) and language models (Bahri et al., 2022). To leverage SAM's generalization ability, Nguyen et al. (2023b) explores the link between flat minima and Bayesian inference, proposing methods to use low-sharpness minima in the Bayesian posterior to enhance neural network generalization. Recently, (Truong et al., 2025b) extends the concept of sharpness to the space of functions that govern the movement of the model particles, and incorporates this theoretical framework to strengthen the generalization ability of the ensemble in Bayesian inference.

### 2.3. Bayesian Approach for Model Fine-tuning

Fine-tuning large architectures like Transformers has increasingly shifted toward parameter-efficient fine-tuning (PEFT) methods such as prompting (Lester et al., 2021), LoRA (Hu et al., 2022), and adapters (Houlsby et al., 2019). These techniques enable efficient adaptation to new tasks without the computational burden of full model retraining. Prompting facilitates rapid task-switching with minimal updates (Lester et al., 2021; Liu et al., 2022), while LoRA (Hu et al., 2022) significantly reduces memory and compute costs by updating only a small subset of parameters. Recently, multiple variants of LoRA have been introduced. For example, Liu et al. (2024) proposes to decompose the weight into the direction and magnitude components and learn these components independently. On the other hand, Truong et al. (2025a) proposes to reparameterize the low-rank matrices with small non-linear networks, hence significantly improving the estimation rate of the low-rank matrices. Besides LoRA, Adapter modules (Sung et al., 2022; Hu et al., 2023;

Zhang et al., 2024) introduce specialized layers that refine model behavior without altering its core weights. Together, these methods make fine-tuning large models more accessible and efficient.

However, despite their efficiency, PEFT methods can lead to overconfident predictions, especially when fine-tuned on small datasets. This has sparked interest in Bayesian approaches for uncertainty-aware adaptation. Several recent advancements integrate Bayesian principles into PEFT to enhance robustness. For example, Bayesian Low-Rank Learning (Doan et al., 2025) applies low-rank perturbations to pre-trained weights, enabling scalable Bayesian neural networks (BNNs), deep ensembles, and Stein Variational Gradient Descent (SVGD) with minimal computational overhead. BayesTune (Kim & Hospedales, 2023) leverages Bayesian inference for more efficient and principled hyperparameter tuning. Laplace-LoRA (Yang et al., 2023) enhances LoRA's calibration by introducing a Laplace approximation, improving uncertainty estimation in large language models (LLMs). Gaussian SWAG + LoRA (Onal et al., 2024) enables lightweight Bayesian inference with negligible computational overhead. Recently, BLoB (Bayesian Low-Rank Adaptation by Backpropagation) (Wang et al., 2024) optimizes both the mean and covariance of LoRA parameters, outperforming post-hoc uncertainty estimation methods. By embedding Bayesian principles into PEFT, these methods not only enhance fine-tuning efficiency but also improve model calibration and uncertainty estimation, addressing a key limitation of prior PEFT approaches.

## 3. Background

### 3.1. Distributional Robustness

This section presents the background on the Wasserstein-based distributional robustness that serves our theory development in the sequel. Distributional robustness (DR) is an emerging framework for learning and decision-making under uncertainty, which seeks the worst-case expected loss among a ball of distributions, containing all distributions that are close to the empirical distribution (Gao et al., 2017).

Consider a generic Polish space $S$ endowed with a distribution $Q$. Let $f : S \to \mathbb{R}$ be a real-valued (risk) function and $c : S \times S \to \mathbb{R}_+$ be a cost function. Distributional robustness setting aims to find the distribution $Q'$ in the vicinity of $Q$ that maximizes the expected risk (Sinha et al., 2018; Blanchet & Murthy, 2019):

$$\max_{Q' : \mathcal{W}_c(Q', Q) < \epsilon} \mathbb{E}_{Q'} [f(z)], \qquad (1)$$

where $\epsilon > 0$ and $\mathcal{W}_c$ denotes the optimal transport (OT) or a Wasserstein distance (Villani, 2008) with respect to the

metric $c$, defined as:

$$\mathcal{W}_c(Q', Q) := \inf_{\gamma \in \Gamma(Q', Q)} \int c \, d\gamma, \qquad (2)$$

where $\Gamma(Q', Q)$ is the set of couplings whose marginals are $Q'$ and $Q$. Under the assumption that $f \in L^1(Q)$ is upper semi-continuous and $c$ is a non-negative lower semi-continuous cost satisfying $c(z, z') = 0$ iff $z = z'$, Blanchet & Murthy (2019) shows that the *dual* form for Eq. (1) is given by:

$$\min_{\lambda \geq 0} \left\{ \lambda \epsilon + \mathbb{E}_{z \sim \mathbb{Q}}[\max_{z'} \{ f(z') - \lambda c(z', z) \}] \right\}. \quad (3)$$

Sinha et al. (2018) further uses a Lagrangian for the Wasserstein-based uncertainty sets to reach a relaxed version with $\lambda \geq 0$:

$$\max_{Q'} \{ \mathbb{E}_{Q'} [f(z)] - \lambda \mathcal{W}_c(Q', Q) \}$$
$$= \mathbb{E}_{z \sim Q}[\max_{z'} \{ f(z') - \lambda c(z', z) \}]. \qquad (4)$$

### 3.2. Fine-tuning Transformer-based Models

Given a transformer-based foundation model $\Phi$, the conventional approach to fine-tuning involves modifying the model's parameters as $\theta = \Phi + \Delta$, where $\Delta$ represents additional modules. Popular parameter-efficient fine-tuning techniques include prompt-tuning (Lester et al., 2021), LoRA (Hu et al., 2022), and Adapters (Houlsby et al., 2019), which are commonly used to adapt models for downstream tasks such as classification on new datasets. To highlight the effectiveness of our Bayesian Inference framework, we focus on the LoRA technique to fine-tune two instances of Transformer-based models, including Vision Transformer (ViT) models (Dosovitskiy et al., 2020) for the image classification task and the Large Language Model LLaMA2 (Touvron et al., 2023) for the commonsense reasoning task.

In brief, LoRA modifies the weight matrix $W$ in the Multi-Head Self-Attention mechanism of the transformer $\Phi$ (e.g., $W^Q$, $W^K$, and $W^V$) by applying the transformation $W \leftarrow W + BA$, where $A$ and $B$ are low-rank matrices. This low-rank reparameterization ensures that the additional parameters $A$ and $B$ remain lightweight, making the approach well-suited for Bayesian inference methods, benefiting from the efficient incorporation of such lightweight modules.

### 3.3. Bayesian Inference with Variational Approach

Consider the model space $\Theta$ over which the parameters $\theta$ follow a prior distribution $P$ with density function $p(\theta)$. Given a training set $\mathcal{S} = \{(x_1, y_1), \dots, (x_N, y_N)\}$ whose examples $(x_i, y_i) \sim \mathcal{D}$, denote $l(\theta; x, y)$ as the loss induced by using the model $\theta$ to predict $x$ with the ground-truth label

$y$ where $l$ is a loss function (e.g., the Cross Entropy (CE) loss). The true posterior is defined as

$$p(\theta \mid \mathcal{S}) \propto \prod_{i=1}^{N} p\left(y_i \mid x_i, \theta\right) p\left(\theta\right),$$

where the likelihood $p(y_i \mid x_i, \theta)$ is defined as

$$p\left(y_i \mid x_i, \theta\right) \propto \exp\left\{-l\left(\theta; x_i, y_i\right)\right\}.$$

Therefore, the true posterior can be rewritten as

$$p(\theta \mid \mathcal{S}) \propto \exp\left\{-\sum_{i=1}^{N} l\left(\theta; x_i, y_i\right)\right\} p\left(\theta\right).$$

Variational approaches (Graves, 2011; Kingma & Welling, 2013; Kingma et al., 2015; Blundell et al., 2015) can be used to obtain an approximate posterior distribution $Q$ with a simpler density function $q(\theta)$ that approximates the true posterior $p(\theta \mid \mathcal{S})$ as:

$$\min_q \left\{ \mathbb{E}_{\theta \sim q} \left[ \sum_{i=1}^{N} l\left(\theta; x_i, y_i\right) \right] + D_{KL}\left(q, p\right) \right\}.$$

Finally, we sample $K$ particle models $\theta_{1:K} \overset{iid}{\sim} Q$ and ensemble their prediction outputs to produce the final result. Evidently, the variational approach for Bayesian inference *lack a mechanism* to explicitly enforce interaction between the particle models $\theta_{1:K}$, such as encouraging these particle models to diverge or complement each other, which is crucial for improving ensemble performance.

Additionally, the Bayesian framework with Stochastic Gradient Langevin Dynamics (SGLD) (Welling & Teh, 2011b) and optimization methods (Nguyen et al., 2023b) encounter the same problem: the particle models $\theta_{1:K} \overset{iid}{\sim} Q$ lack a mechanism to encourage the particle models to diverge or complement each other.

# 4. Interactive Bayesian Distributional Robustness

## 4.1. Motivations

As discussed earlier, promoting diversity among particles is crucial to prevent mode collapse. However, conventional Bayesian frameworks lack explicit mechanisms to model interactions between particles $\theta_{1:K}$ during training. To address this, we first define the **approximate posterior distribution** $Q^K$ over the product space $\Theta^K$, where samples are concatenated particle models, $\boldsymbol{\theta} = \theta_{1:K}$. This formulation allows us to define a loss function over the joint particle models, thereby facilitating the modeling of the interactions between the individual particles $\theta_{1:K}$.

Our framework learns $Q$ such that if $\boldsymbol{\theta} = \theta_{1:K} \sim Q^K$ or $\theta_{1:K} \overset{iid}{\sim} Q$, the models reside in *low-loss, low-sharpness* regions while *maintaining diversity*. To achieve this, we introduce a novel loss function that encourages particle interaction during training, guiding them toward regions of low sharpness while maintaining high diversity. Then, the i.i.d samples (i.e., the models) sampled from this posterior will also yield high diversity and low sharpness. Notably, particles interact only during training to shape the final posterior $Q$, but they are sampled independently at inference.

One potential drawback of promoting particle interactions is the risk of uncontrolled instabilities during training. To alleviate this issue and enhance robustness within this interactive framework, we employ WS-based Distributional Robustness Optimization (DRO) to develop Theorem 4.1. This theorem characterizes the population loss over the approximate posterior $Q^K$, providing insights into a practical method for promoting particle diversity and improving generalization. Specifically, Theorem 4.1 can be interpreted as finding particle models that exhibit both **diversity** and **low sharpness**, enhancing their ability to generalize, as discussed in prior works including Sharpness-Aware Minimization (SAM) (Foret et al., 2021). We emphasize that this result is a generalization upon prior findings on distributional robustness, as elaborated in Corollary 4.2.

## 4.2. Theoretical Development

Let $\Theta$ be the model space and $Q$ and $P$ be the approximate and prior distributions over $\Theta$. Given a positive number of particle models $K > 0$. To be able to model the interactions between these $K$ particles, we first define $Q^K = \underbrace{Q \odot Q \odot \cdots \odot Q}_{K \text{ times}}$ and $P^K = \underbrace{P \odot P \odot \cdots \odot P}_{K \text{ times}}$ as the joint distributions of $K$ statistically independent particle models sampled from $Q$ and $P$, respectively. Denote $\boldsymbol{\theta} = \theta_{1:K} \sim Q^K$ as the concatenation of the $K$ models, we propose the following loss function

$$\ell(\boldsymbol{\theta}; x, y) = \frac{1}{K} \sum_{i=1}^{K} l(\theta_i; x, y) + \alpha l_{div}(\theta_{1:K}; x, y),$$

where $(x, y) \sim \mathcal{D}$ is sampled from the data distribution (i.e., $\mathcal{D}$ is the general distribution of data/label pairs), $l$ is the loss function (e.g., CE loss or Hinge loss), $l_{div}$ is the *divergence loss* which encourages the particles $\theta_{1:K}$ to be diverse and will be explicitly defined later, and $\alpha > 0$ is a trade-off hyperparameter capturing the extent to which we want to encourage the diversity between the particles.

Given a training set $\mathcal{S} = \{(x_i, y_i)\}_{i=1}^{N} \sim \mathcal{D}^N$, we define the following population loss functions over a single model

$\boldsymbol{\theta}$ and over the approximate posterior $Q^K$ as:

$$\mathcal{L}_{\mathcal{D}}(\boldsymbol{\theta}) = \mathbb{E}_{(x,y)\sim\mathcal{D}}\Big[\ell(\boldsymbol{\theta}; x, y)\Big]$$

$$\mathcal{L}_{\mathcal{D}}\Big(Q^K\Big) = \mathbb{E}_{\boldsymbol{\theta}\sim Q^K}\Big[\mathcal{L}_{\mathcal{D}}\Big(\boldsymbol{\theta}\Big)\Big].$$

Similarly, we define the empirical losses over a single model and over the approximate posterior $Q^K$ as:

$$\mathcal{L}_S(\boldsymbol{\theta}) = \mathbb{E}_{(x,y)\sim S}\Big[\ell(\boldsymbol{\theta}; x, y)\Big],$$

$$\mathcal{L}_{\mathcal{S}}\Big(Q^K\Big) = \mathbb{E}_{\boldsymbol{\theta}\sim Q^K}\Big[\mathcal{L}_{\mathcal{S}}\Big(\boldsymbol{\theta}\Big)\Big].$$

Note that the population loss $\mathcal{L}_{\mathcal{D}}\Big(Q^K\Big)$ takes the form:

$$\mathcal{L}_{\mathcal{D}}\Big(Q^K\Big) = \mathbb{E}_{\boldsymbol{\theta}\sim Q^K}\Big[\mathcal{L}_{\mathcal{D}}\Big(\boldsymbol{\theta}\Big)\Big]$$

$$= \mathbb{E}_{\boldsymbol{\theta}\sim Q^K}\Big[\frac{1}{K}\sum_{i=1}^{K}\mathcal{L}_{\mathcal{D}}(\theta_i) + \alpha\mathbb{E}_{\mathcal{D}}\Big[l_{div}(\theta_{1:K}; x, y)\Big]\Big].$$

By minimizing the general loss $\mathcal{L}_{\mathcal{D}}(Q^K)$, we simultaneously encourage the particle models to achieve *low general loss* while *diverging* from each other to enhance ensemble performance. A key challenge is that directly minimizing the general loss $\mathcal{L}_{\mathcal{D}}(Q^K)$ is impractical since we do not have access to the data distribution $\mathcal{D}$. To address this challenge, we present the following theorem, whose proof can be found in Appendix C.

**Theorem 4.1.** *With the probability at least $1 - \delta$ over the choice of $\mathcal{S} \sim \mathcal{D}^N$, we have*

$$\mathcal{L}_{\mathcal{D}}\Big(Q^K\Big) \le L\sqrt{\frac{KD_{KL}\Big(Q, P\Big) + \log\frac{1}{\delta}}{2N}}$$

$$+ \min_{\lambda \ge 0}\Big\{\lambda\rho + \mathbb{E}_{\boldsymbol{\theta}\sim Q^K}\Big[\max_{\boldsymbol{\theta}'}\Big\{\mathcal{L}_{\mathcal{S}}(\boldsymbol{\theta}') - \lambda c^K(\boldsymbol{\theta}, \boldsymbol{\theta}')\Big\}\Big]\Big\},$$

*where $\mathcal{D}$ is the general data/label distribution, $L$ is the upper-bound of the loss function $\ell(\boldsymbol{\theta}; x, y)$, and $c^K(\boldsymbol{\theta}, \boldsymbol{\theta}') = \frac{1}{K}\sum_{i=1}^{K}c(\theta_i, \theta_i')$ represents a distance/divergence between two models.*

Theorem 4.1 offers a practical alternative, as it provides an upper bound involving empirical losses over the training set $S$, which can be computed given the availability of the training set $\mathcal{S}$. This upper bound is appealing due to its connection with Sharpness-Aware Minimization (SAM) (Foret et al., 2021). Specifically, in the inner maximization, the parameter $\lambda$ controls the distance between the adversarial model $\boldsymbol{\theta}'$ and the center model $\boldsymbol{\theta}$. The outer minimization balances the terms $\lambda\rho$ and $\mathbb{E}_{\boldsymbol{\theta}\sim Q^K}\Big[\max_{\boldsymbol{\theta}'}\Big\{\mathcal{L}_{\mathcal{S}}(\boldsymbol{\theta}') - \lambda c(\boldsymbol{\theta}, \boldsymbol{\theta}')\Big\}\Big]$, where an increase in $\lambda$ raises the first term but reduces the second.

To further illustrate the connection to SAM, we present the following corollary, where we adopt a specific form of the cost function $c$, establishing a clear link SAM (Foret et al., 2021). The proof can be found in Appendix C.

**Corollary 4.2.** *Given a metric $d$ over the model space, consider the following cost function $c$*

$$c(\boldsymbol{\theta}, \boldsymbol{\theta}') = \begin{cases} d(\boldsymbol{\theta}, \boldsymbol{\theta}') & \text{if } d(\boldsymbol{\theta}, \boldsymbol{\theta}') \le \rho \\ +\infty & \text{otherwise.} \end{cases}$$

*With the prob. at least $1 - \delta$ over the choice of $\mathcal{S} \sim \mathcal{D}^N$,*

$$\mathcal{L}_{\mathcal{D}}\Big(Q^K\Big) \le \mathbb{E}_{\boldsymbol{\theta}\sim Q^K}\Big[\max_{\boldsymbol{\theta}'\in\mathcal{B}_\rho(\boldsymbol{\theta})}\mathcal{L}_{\mathcal{S}}\Big(\boldsymbol{\theta}'\Big)\Big]$$

$$+ L\sqrt{\frac{KD_{KL}\Big(Q, P\Big) + \log\frac{1}{\delta}}{2N}},$$

*where the ball $\mathcal{B}_\rho(\boldsymbol{\theta}) := \{\boldsymbol{\theta}' : d(\boldsymbol{\theta}, \boldsymbol{\theta}') \le \rho\}$.*

If we further simplify the analysis by considering the $l_2$ Euclidean distance for $d$, Corollary 4.2 reveals that the Sharpness-Aware Bayesian Neural Network (SA-BNN) framework from (Nguyen et al., 2023b) becomes a special case of our broader approach. However, compared to SA-BNN, our method, as established in Theorem 4.1 and Corollary 4.2, offers a notable advancement: *this framework operates on the joint distribution $Q^K$ and incorporates a divergence loss $l_{div}$, enabling us to model interactions between the particle models $\theta_{1:K}$*. This inter-particle interaction is beneficial to achieve strong ensemble accuracy, as we will empirically demonstrate in the subsequent experiments.

### 4.3. Practical Algorithm

This section explores the theory above to derive a practical method Interactive Bayesian Distribution Robustness for Model Fine-tuning (IBDR). We first discuss how to model the divergence loss $l_{div}(\theta_{1:K}; x, y)$. Given a pair $(x, y) \in S$, denote $f(x; \theta_i)$ as the prediction probabilities of the particle model $\theta_i$ on $x$. Let $f_{-y}(x; \theta_i)$ (or $f_{-y}^i$ for short, when the context is clear) be the *non-maximal* prediction probabilities by eliminating the prediction probability of the ground-truth label $y$. Inspired by Pang et al. (2019), we encourage the non-maximal predictions $f_{-y}^i$ ($i = 1, \ldots, K$, where $K$ is the number of particle models) to diverge, while maximizing $f_y^i$ ($i = 1, \ldots, K$). Motivated by the theory of Determinantal Point Processes (DPP) (Kulesza et al., 2012), the ensemble diversity can be defined as:

$$l_{div}\Big(\theta_{1:K}; x, y\Big) = \det\Big(\Big[\tilde{f}_{-y}^i\Big]_{i\in[K]}^T\Big[\tilde{f}_{-y}^i\Big]_{i\in[K]}\Big),$$

where $\tilde{f}_{-y}^i = \frac{f_{-y}^i}{\|f_{-y}^i\|}$ and $\Big[\tilde{f}_{-y}^i\Big]_{i\in[K]} \in \mathbb{R}^{(C-1)\times K}$, $\Big[K\Big] = \Big\{1, ..., K\Big\}$ and C is the number of classes.

Moreover, according to the matrix theory (Bernstein, 2009),

$$\det\left(\left[\tilde{f}^i_{-y}\right]^T_{i\in[K]}\left[\tilde{f}^i_{-y}\right]_{i\in[K]}\right) = \text{Vol}^2\left(\left[\tilde{f}^i_{-y}\right]_{i\in[K]}\right),$$

where $\text{Vol}\left(\left[\tilde{f}^i_{-y}\right]_{i\in[K]}\right)$ specifies the volume spanned the vectors in $\left[\tilde{f}^i_{-y}\right]_{i\in[K]}$, indicating that we aim to maximize the diversity of the non-maximal predictions by maximally increasing their spanned volume.

We define the approximate posterior distribution as a mixture of Gaussians: $Q = \frac{1}{K}\sum_{i=1}^K \mathcal{N}\left(\mu_i, \sigma^2\mathbb{I}\right)$, with the prior distribution given by $P = \mathcal{N}\left(\mathbf{0}, \mathbb{I}\right)$. Using this setup, we have the following corollary, which provides valuable insights for developing a practical method. The proof of this corollary can be found in Appendix C.

**Corollary 4.3.** *With the probability at least $1 - \delta$ over the choice of $\mathcal{S} \sim \mathcal{D}^N$, we have*

$$\mathcal{L}_{\mathcal{D}}\left(Q^K\right) \leq \min_{\lambda \geq 0}\left\{\lambda\rho + \mathbb{E}_{\theta_{1:K}\sim Q}\left[\max_{\theta'_{1:K}}\left\{\frac{\sum_{i=1}^K l(\theta'_i; x, y)}{K}\right.\right.\right.$$

$$\left.\left.\left. + \alpha l_{div}(\theta'_{1:K}; x, y) - \frac{\lambda}{K}\sum_{i=1}^K c(\theta_i, \theta'_i)\right\}\right]\right\}$$

$$+ L\sqrt{\frac{\sum_{i=1}^K \|\mu_i\|^2 + Kd(\sigma - \log\sigma) + 2\log\frac{1}{\delta}}{4N}},$$

*where $\mathcal{D}$ is the general data/label distribution, $L$ is the upper-bound of the loss $\ell$, and $d$ is the model size.*

Inspired by this corollary, we apply a minor relaxation to the right-hand side and propose to solve the following minimization problem:

$$\min_{\mu_{1:K},\sigma} \min_{\lambda \geq 0}\left\{\lambda\rho + \mathbb{E}_{\theta_{1:K}\sim Q}\left[\frac{1}{K}\sum_{i=1}^K \max_{\theta'_i}\tilde{\ell}(\theta'_i, \theta_i; x, y)\right]\right\}$$

$$+ \frac{\beta}{K}\left[\sum_{i=1}^K \|\mu_i\|^2 + d(\sigma - \log\sigma)\right], \quad (5)$$

where $\tilde{\ell}(\theta'_i, \theta_i; x, y) = l(\theta'_i; x, y) + \alpha l_{div}(\theta'_i, \theta_{-i}; x, y) - \lambda c(\theta_i, \theta'_i)$. Here, $\beta$ can be interpreted as a regularization hyper-parameter. We denote the loss function in Eq. (5) as $\overline{\mathcal{L}}(\lambda, \theta'_i, \theta_i; x, y)$. We employ the reparameterization trick, expressing $\theta_i = \mu_i + \sigma\epsilon_i$ with $\epsilon_i \sim \mathcal{N}\left(\mathbf{0}, \mathbb{I}\right)$. Additionally, for each $\theta'_i$ we relax the divergence loss $l_{div}\left(\theta'_{1:K}; x, y\right)$ to $l_{div}\left(\theta'_i, \theta_{-i}; x, y\right)$, where $\theta_{-i} = \left[\theta_j\right]_{j\neq i}$.

**Practical algorithm.** To solve the optimization problem in Eq. (5), we alternatively update $\mu_{1:K}$ and $\lambda$ using gradient

---

**Algorithm 1** **I**nteractive **B**ayesian **D**istributional **R**obustness (IBDR)

> **Input:** Initial particle means $\mu_{1:K}$; ascend step size $\alpha_1$; learning rates $\alpha_\lambda, \alpha_\mu$
> **Output:** Optimal particle means $\mu_{1:K}$
> **while** not converged **do**
>   Sample batch $\mathcal{B} = \{(x_1, y_1), \ldots, (x_b, y_b)\}$
>   Sample $\epsilon_i \sim \mathbb{N}(0, \mathbb{I})$ and $\theta_i \leftarrow \mu_i + \sigma\epsilon_i$
>   Compute $\theta'_i \leftarrow \theta_i + \alpha_1 \nabla_{\theta_i}\tilde{\ell}(\theta'_i, \theta_i; x, y)$
>   Compute $\lambda \leftarrow \lambda - \alpha_\lambda \nabla_\lambda \overline{\mathcal{L}}(\lambda, \theta'_i, \theta_i; x, y)$
>   Compute $\mu_i \leftarrow \lambda - \alpha_\mu \nabla_{\mu_i}\overline{\mathcal{L}}(\lambda, \theta'_i, \theta_i; x, y)$
> **end while**
> **return** $\mu_{1:K}$

---

descent. In practice, we fix $\sigma = 0.1$ and do not learn $\sigma$. To update $\mu_{1:K}$, we first apply the reparameterization trick to obtain $\theta_{1:K}$ and use one-step gradient ascend to obtain $\theta'_{1:K}$. We next consider $\theta'_i = \theta'_i(\theta_i) = \theta'_i(\mu_i + \sigma\epsilon_i)$ and apply the chain rule to compute the gradient of the loss w.r.t. $\mu_i$. Similar to SAM, we simplify the derivative of $\theta'_i$ w.r.t. $\theta_i$ as the identity matrix, hence leading to the gradient of the loss w.r.t. $\mu_i$ is that of the loss w.r.t. $\theta'_i$. Finally, we apply one-step projected gradient descent (e.g., to ensure $\lambda \geq 0$) to update $\lambda$. The pseudo-code of our approach is summarized in Algorithm 1.

## 5. Experiments

We focus on the fine-tuning problem, where a pre-trained model, denoted as $\Phi$, is provided, and the goal is to identify the optimal parameters $\theta = \Phi + \Delta$, with $\Delta$ representing an additional component. Various parameter-efficient fine-tuning (PEFT) methods, such as LoRA (Hu et al., 2022) and Adapters (Houlsby et al., 2019), have been developed to achieve this objective and have demonstrated remarkable performance compared to the conventional full fine-tuning. Since $\Delta$ is typically a much smaller component than the complete model in PEFT methods, Bayesian techniques offer promising applications in model fine-tuning. To assess the versatility and effectiveness of our method, we experiment with two tasks on different domains: **image classification** and **commonsense reasoning**.

### 5.1. Image Classification

We fine-tuned the ViT-B/16 architecture, pre-trained on the `ImageNet-21K` dataset (Deng et al., 2009), using the LoRA framework to learn the parameters $\Delta$. Within the Bayesian inference framework, our goal is to learn $K$ LoRA particles $\Delta_i$, each producing a unique model instance $\theta_i$. The final output is then generated by averaging the predictions from these model instances.

To evaluate the performance of IBDR, we conducted experi-

*Table 1.* Top-1 Accuracy on VTAB-1K. The accuracies are reported with ViT-B/16 pre-trained on ImageNet-21K

| Method | Natural | | | | | | | Specialized | | | | Structured | | | | | | | | AVG |
|---|---|---|---|---|---|---|---|---|---|---|---|---|---|---|---|---|---|---|---|---|
| | CIFAR100 | Caltech101 | DTD | Flowers102 | Pets | SVHN | Sun397 | Camelyon | EuroSAT | Resisc45 | Retinopathy | Clevr-Count | Clevr-Dist | DMLab | KITTI | dSpr-Loc | dSpr-Ori | sNORB-Azim | sNORB-Ele | |
| FFT | 68.9 | 87.7 | 64.3 | 97.2 | 86.9 | 87.4 | 38.8 | 79.7 | **95.7** | 84.2 | 73.9 | 56.3 | 58.6 | 41.7 | 65.5 | 57.5 | 46.7 | 25.7 | 29.1 | 62.3 |
| LoRA | 67.1 | 90.7 | 68.9 | 98.1 | 90.1 | 84.5 | 54.2 | 84.1 | 94.9 | 84.4 | 73.6 | **82.9** | 69.2 | 49.8 | 78.5 | 75.7 | 47.1 | **31.0** | **44.0** | 68.4 |
| SAM | 72.7 | 90.3 | 71.4 | 99.0 | 90.2 | 84.4 | 52.4 | 82.0 | 92.6 | 84.1 | 74.0 | 76.7 | 68.3 | 47.9 | 74.3 | 71.6 | 43.4 | 26.9 | 39.1 | 70.5 |
| SA-BNN | 65.1 | 91.5 | 71.0 | 98.9 | 89.4 | 89.3 | 55.2 | **86.2** | 94.5 | 86.4 | 75.2 | 61.4 | 63.2 | 40.0 | 71.3 | 64.5 | 34.5 | 27.2 | 31.2 | 68.2 |
| SGLD | 68.7 | 91.0 | 67.0 | 98.6 | 89.3 | 83.0 | 51.6 | 81.2 | 93.7 | 83.2 | 76.4 | 80.0 | 70.1 | 48.2 | 76.2 | 71.1 | 39.3 | 31.2 | 38.4 | 68.4 |
| DeepEns | 68.6 | 88.9 | 67.7 | 98.9 | 90.7 | 85.1 | 54.5 | 82.6 | 94.8 | 82.7 | 75.3 | 46.6 | 47.1 | 47.4 | 68.2 | 71.1 | 36.6 | 30.1 | 35.6 | 67.0 |
| BayesTune | 68.2 | 91.7 | 69.5 | 99.0 | 90.2 | 86.4 | 51.2 | 84.9 | 95.3 | 84.1 | 75.1 | 82.8 | 68.9 | 49.7 | 79.3 | 74.3 | 46.6 | 30.3 | 42.8 | 68.5 |
| SVGD | 71.3 | 90.2 | 71.0 | 98.7 | 90.2 | 84.3 | 52.7 | 83.4 | 93.2 | 86.7 | 75.1 | 75.8 | 70.7 | 49.6 | 79.9 | 69.1 | 41.2 | 30.6 | 33.1 | 70.9 |
| IBDR | **73.0** | **92.1** | **71.7** | **99.3** | **91.4** | **91.3** | **56.7** | 85.1 | 95.0 | **87.3** | **76.5** | 78.1 | **75.1** | **53.6** | **80.4** | **77.1** | **49.3** | 28.9 | 40.1 | **73.6** |
| | (.11) | (.31) | (.12) | (0.15) | (0.16) | (.36) | (.18) | (.24) | (.44) | (.14) | (.12) | (.11) | (.24) | (.42) | (.26) | (.29) | (.19) | (.13) | (.37) | |

*Table 2.* Expected Calibration Errors (ECE) on `VTAB-1K`. The results are reported with `ViT-B/16` pre-trained on `ImageNet-21K`

| Method | Natural | | | | | | | Specialized | | | | Structured | | | | | | | | AVG |
|---|---|---|---|---|---|---|---|---|---|---|---|---|---|---|---|---|---|---|---|---|
| | CIFAR100 | Caltech101 | DTD | Flowers102 | Pets | SVHN | Sun397 | Camelyon | EuroSAT | Resisc45 | Retinopathy | Clevr-Count | Clevr-Dist | DMLab | KITTI | dSpr-Loc | dSpr-Ori | sNORB-Azim | sNORB-Ele | |
| FFT | 0.29 | 0.23 | 0.20 | 0.13 | 0.27 | 0.19 | 0.45 | 0.21 | 0.13 | 0.18 | 0.17 | 0.41 | 0.44 | 0.42 | 0.22 | 0.14 | 0.23 | 0.24 | 0.40 | 0.26 |
| LoRA | 0.38 | 0.19 | 0.18 | 0.05 | 0.09 | 0.10 | 0.14 | **0.11** | 0.09 | 0.12 | 0.11 | 0.12 | 0.19 | 0.34 | 0.18 | 0.14 | 0.21 | 0.18 | 0.31 | 0.17 |
| SAM | 0.21 | 0.25 | 0.20 | 0.11 | 0.12 | 0.15 | 0.14 | 0.17 | 0.16 | 0.14 | **0.09** | 0.12 | 0.17 | 0.24 | 0.16 | 0.21 | **0.19** | 0.13 | 0.16 | 0.16 |
| SA-BNN | 0.22 | 0.08 | 0.19 | 0.15 | 0.12 | 0.12 | 0.24 | 0.13 | **0.06** | 0.12 | 0.18 | 0.14 | **0.21** | 0.22 | 0.24 | 0.25 | 0.41 | 0.46 | 0.34 | 0.20 |
| SGLD | 0.26 | 0.20 | **0.17** | 0.05 | 0.18 | 0.14 | 0.23 | 0.18 | 0.09 | 0.12 | 0.32 | 0.26 | 0.29 | **0.21** | 0.26 | 0.42 | 0.39 | **0.11** | 0.24 | 0.22 |
| DeepEns | 0.24 | 0.12 | 0.22 | 0.04 | 0.10 | 0.13 | 0.23 | 0.16 | 0.07 | 0.15 | 0.21 | 0.31 | 0.32 | 0.36 | 0.13 | 0.32 | 0.31 | 0.16 | 0.29 | 0.20 |
| BayesTune | 0.32 | 0.93 | 0.20 | 0.03 | 0.85 | 0.12 | 0.22 | 0.13 | 0.07 | 0.13 | 0.22 | **0.12** | 0.23 | 0.30 | 0.24 | 0.28 | 0.28 | 0.31 | 0.26 | 0.23 |
| SVGD | 0.20 | 0.13 | 0.19 | 0.04 | 0.16 | 0.09 | 0.20 | 0.15 | 0.11 | 0.13 | 0.12 | 0.17 | 0.21 | 0.30 | 0.18 | 0.21 | 0.25 | 0.14 | 0.26 | 0.18 |
| IBDR | **0.16** | **0.08** | 0.19 | **0.02** | **0.07** | **0.07** | **0.13** | 0.12 | 0.06 | **0.11** | 0.11 | 0.13 | 0.24 | 0.30 | **0.12** | **0.11** | 0.30 | 0.30 | **0.16** | **0.14** |
| | (.03) | (.02) | (.02) | (.01) | (.01) | (.01) | (.02) | (.03) | (.02) | (.02) | (.01) | (.01) | (.02) | (.03) | (.01) | (.01) | (.05) | (.04) | (.02) | |

*Table 3.* Accuracy/ECE on six common-sense reasoning datasets

| Metric | | Datasets | | | | | | |
|---|---|---|---|---|---|---|---|---|
| Type | Method | WG-S | ARC-C | ARC-E | WG-M | OBQA | BoolQ | AVG |
| ACC (↑) | MLE | 68.99 | 69.10 | 85.65 | 74.53 | 81.52 | 86.53 | 77.72 |
| | MAP | 68.62 | 67.59 | 86.55 | 75.61 | 81.38 | 86.50 | 77.71 |
| | MCD | 69.26 | 68.43 | 86.07 | 76.18 | 81.49 | 87.15 | 78.10 |
| | ENS | 69.57 | 66.20 | 84.40 | 75.32 | 81.38 | 87.09 | 77.33 |
| | BBB | 67.54 | 68.11 | 85.63 | 73.41 | 81.72 | **87.19** | 77.27 |
| | LAP | 69.20 | 66.78 | 80.05 | 75.55 | 82.12 | 86.95 | 76.78 |
| | BLoB | 70.89 | **70.83** | 86.68 | 74.55 | 82.73 | 86.80 | 78.75 |
| | IBDR | **72.51** | 70.56 | **86.95** | **76.46** | **84.60** | 86.89 | **79.66** |
| ECE (↓) | MLE | 29.83 | 29.00 | 13.12 | 20.62 | 12.55 | 3.18 | 18.05 |
| | MAP | 29.76 | 29.42 | 12.07 | 23.07 | 13.26 | 3.16 | 18.46 |
| | MCD | 28.06 | 27.73 | 12.31 | 18.27 | 15.12 | 3.49 | 17.50 |
| | ENS | 28.52 | 29.16 | 12.57 | 20.86 | 15.34 | 9.61 | 19.34 |
| | BBB | 21.93 | 25.84 | 12.42 | 15.89 | 11.23 | 3.76 | 15.18 |
| | LAP | **4.15** | **16.25** | 33.29 | **7.40** | 8.70 | **1.30** | **11.85** |
| | BLoB | 20.62 | 20.61 | **9.43** | 11.23 | 8.36 | 2.46 | 12.12 |
| | IBDR | 24.17 | 21.20 | 9.71 | 11.19 | **5.82** | 1.54 | 12.27 |

ments using the `VTAB-1K` benchmark (Zhai et al., 2020), a standardized framework designed to assess the transfer learning capabilities of models across a diverse range of visual tasks. This benchmark includes 19 distinct datasets encompassing three domains: Natural images, Specialized, and Structured. Each task in `VTAB-1K` is constrained to 1,000 labeled examples for fine-tuning, presenting a challenging scenario for evaluating the model's ability to generalize across domains with limited data.

We benchmarked IBDR against 8 baselines, utilizing three deterministic fine-tuning approaches: full fine-tuning (FFT), AdamW, and SAM, as well as five Bayesian inference methods: Sharpness-Aware Bayesian Neural Network (SA-BNN) (Nguyen et al., 2023b), Stochastic Gradient Langevin Dynamics (SGLD) (Welling & Teh, 2011a), Bayesian Deep Ensembles (DeepEns) (Lakshminarayanan et al., 2017), BayesTune (Kim & Hospedales, 2023) and Stein Variational Gradient Descent (SVGD) (Liu & Wang, 2016).

In our experiment, we set $\alpha = 0.02, \beta = 10^{-4}$ for all datasets. We tune $\rho$ using the provided validation set, with the candidate set being $\rho \in \{0.01, 0.05, 0.1\}$. All models, except for the deterministic methods (including SAM, LoRA with the AdamW optimizer, and full fine-tuning), were trained using *four particles* on the same set of LoRA parameters specified by Hu et al. (2022). We conducted three independent runs for each experiment and reported the mean scores and standard deviation. For additional information regarding the experimental setup, please refer to Appendix A.

According to Table 1, IBDR outperforms all baselines by large margins. We also note that IBDR surpasses SA-BNN by more than 5%, underscoring the effectiveness of incorporating inter-particle interaction for ensemble diversity. While the inter-particle interactions can enhance ensemble

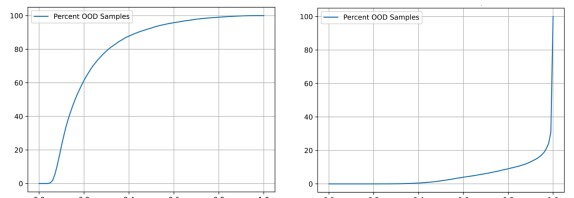

*Figure 1.* Percentage of OOD samples at different thresholds. Left: model trained on CIFAR-100 and tested on SVHN. Right: trained and tested on SVHN.

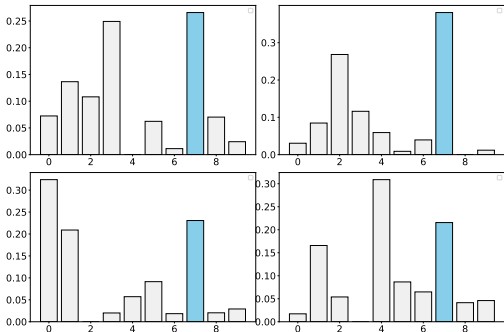

*Figure 2.* Average prediction probabilities of four particles on the SVHN testing set. The blue bar indicates the ground truth label.

diversity, these repulsive forces may compromise model robustness. However, our approach mitigates this issue by incorporating distributional robustness into the interactive framework. To evaluate the robustness of IBDR, we report the Expected Calibration Error (ECE) in Table 2. Although there is often a trade-off between accuracy and ECE, IBDR maintains a good balance between these metrics and achieves the best ECE among the baselines. This result highlights the effectiveness of our method in balancing distributional robustness and particle diversity.

## 5.2. Commonsense Reasoning

Having shown that IBDR excels on the vision task, we extend our experiment to the commonsense reasoning task. We fine-tuned the LLaMA2-7B model on six widely-used common-sense reasoning tasks: ARC-Challenge (ARC-C) and ARC-Easy (ARC-E) (Clark et al., 2018), Winogrande-Small (WG-S) and Winogrande-Medium (WG-M) (Sakaguchi et al., 2021), OpenBookQA (OBQA) (Mihaylov et al., 2018), and BoolQ (Clark et al., 2019). Following the experimental setup outlined in Wang et al. (2024), we benchmarked IBDR against seven baseline methods, including Maximum Likelihood Estimation (MLE), Maximum A Posteriori (MAP), Monte Carlo Dropout (MCD) (Gal & Ghahramani, 2016), Bayes By Backprop (BBB) (Blundell et al., 2015), Deep Ensembles (ENS) (Lakshminarayanan et al., 2017), the recently proposed LaplaceLoRA (LAP) (Yang et al., 2023) and Bayesian Low-Rank Adaptation by Backpropagation (BLoB) (Wang et al., 2024). For all baseline methods, we used the same frozen pre-trained LLM backbone. To maintain consistency, we kept hyperparameters identical across all datasets. Except for a few IBDR-specific hyperparameters, we strictly adhered to the default settings from Hu et al. (2022).

The accuracy and Expected Calibration Error (ECE) metrics are presented in Table 3, demonstrating that IBDR outperforms the baseline methods across most tasks, achieving notable performance improvement while maintaining competitive calibration. Also, IBDR's ability to balance predictive accuracy with uncertainty calibration highlights its effectiveness in modeling inter-particle interactions.

## 6. Ablation Studies

### 6.1. Out-of-Distribution Performance

We conducted additional experiments to evaluate our model's out-of-distribution (OOD) detection capability. As shown in Figure 1, we plotted two line graphs illustrating the percentages of OOD samples at corresponding thresholds. The left graph represents the results for a model trained on the CIFAR-100 dataset and tested on the SVHN dataset, while the right graph illustrates the results for a model both trained and tested on the SVHN dataset. As expected, when evaluated on a dataset different from the training data, our model exhibits low confidence across all classes and successfully identifies these samples as OOD. In contrast, the results in the right graph demonstrate that the model trained and tested on the SVHN dataset achieves high confidence in its predictions, further supporting our findings.

### 6.2. Particle Interactions and Ensemble Diversity

This section delves into the effectiveness of our framework in enhancing ensemble diversity. As shown in Figure 2, which visualizes the average prediction probabilities of four particles, two out of the four particles assign the highest probability to the ground truth class, while the other two predict the ground truth as one of the top two probabilities. As discussed in Section 4.3, our algorithm is designed to promote *diversity in nonmaximal probability predictions*. This is evident in Figure 2. Indeed, the third particle predicts class 0 with high probability. Through interaction among the particles, the other particles tend to assign lower probabilities to class 0, and this effect extends to all classes that are not the ground truth. Then, when taking the average of the predictions across all particles, the overall probability for non-ground-truth classes is reduced, leading to the dominant probability of 30% for the ground truth compared to the probability of less than 15% for any other class in this case. By fostering particle diversity and maximizing the spanned volume of nonmaximal predictions, we decreased

the chance of multiple particles making the same misprediction and enhanced the ensemble quality.

## 7. Conclusion

We present Interactive Bayesian Distributional Robustness (IBDR), a novel Bayesian inference framework that models interactions between particles to simultaneously enhance ensemble diversity and distributional robustness. IBDR has been rigorously tested on different tasks from various domains, where it notably outperformed the baselines, demonstrating its effectiveness across diverse tasks.

## Acknowledgments

Trung Le was partly supported by ARC DP23 grant DP230101176 and by the Air Force Office of Scientific Research under award number FA9550-23-S-0001.

## Impact Statement

This paper presents work whose goal is to advance the field of Machine Learning. There are potential societal consequences of our work, none of which we feel must be specifically highlighted here.

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

# Supplement for "Promoting Ensemble Diversity with Interactive Bayesian Distributional Robustness for Fine-tuning Foundation Models"

The Supplementary Material is structured as follows: Appendix A presents the additional experiments to demonstrate the effectiveness and robustness of our method. Appendix B discusses the experimental information, including hyperparameters choice and the details about datasets. Appendix C presents the proofs of the theoretical results.

## A. Additional Experiments

To further evaluate the performance and behavior of IBDR, we conduct additional experiments in this section. First, we examine the impact of the number of particles, verifying the benefits of using multiple diverse particles and supporting our choice for the optimal number. Additionally, we assess IBDR's performance across various hyperparameter values, demonstrating its robustness with respect to different settings of $\rho$ and $\alpha$.

### A.1. Effect of Number of particles

We conducted experiments on the four Specialized datasets, including the Patch Camelyon, EuroSAT, Resics45, and Diabetic Retinopathy datasets, to assess how the number of particles affects final performance. As shown in Table 4, increasing the number of particles enhances the ensemble's quality, improving performance. However, Table 5 shows that increasing the number of particles also results in a linear increase in runtime. Considering this tradeoff between time and memory, we determined that using #PARTICLES $= 4$ offers an optimal balance between accuracy and training cost.

*Table 4.* Classification accuracy with different #PARTICLES

| #PARTICLES | Camelyon | EuroSAT | Resics45 | Retinopathy |
|:---:|:---:|:---:|:---:|:---:|
| 1p | 82.4 | 93.1 | 84.2 | 73.8 |
| 2p | 84.8 | 93.9 | 86.6 | 74.3 |
| 4p | 85.1 | 95.0 | 87.3 | 76.5 |
| 8p | 85.8 | 95.5 | 87.4 | 77.0 |

*Table 5.* Runtime per epoch with different #PARTICLES

| #PARTICLES | Camelyon | EuroSAT | Resics45 | Retinopathy |
|:---:|:---:|:---:|:---:|:---:|
| 1p | $51_{\pm1.8}$ | $50_{\pm1.5}$ | $48_{\pm1.7}$ | $51_{\pm0.7}$ |
| 2p | $80_{\pm2.1}$ | $83_{\pm2.4}$ | $93_{\pm2.1}$ | $85_{\pm0.9}$ |
| 4p | $158_{\pm4.3}$ | $161_{\pm4.9}$ | $156_{\pm4.1}$ | $151_{\pm2.1}$ |
| 8p | $220_{\pm6.2}$ | $230_{\pm5.7}$ | $218_{\pm7.3}$ | $246_{\pm6.8}$ |

### A.2. Effect of particle interaction via $l_{div}$

To further assess the impact of particle interactions, we examine the effect of varying values of $\alpha$, which indicates the extent to which we enforce diversity among particles through the divergence loss $l_{div}$. There is a significant performance gap between $\alpha = 0$ and $\alpha = 0.02$ on all datasets, indicating the importance of the divergence loss and particle diversity. As $\alpha$ increases, we anticipate that the behavior of the particles may become increasingly unstable, as the divergence loss begins to

*Table 6.* Accuracy with different values of $\alpha$ on DTD, DMLab, and SVHN datasets

| $\alpha$ | DTD | DMLAB | SVHN |
|---|---|---|---|
| 0 | 67.24 | 51.92 | 87.03 |
| 0.02 | 71.70 | **53.61** | **91.32** |
| 0.08 | **71.73** | 52.73 | 91.26 |
| 0.5 | 70.86 | 53.16 | 90.63 |
| 1.5 | 70.93 | 51.27 | 90.34 |
| 3 | 68.32 | 48.64 | 86.31 |

*Table 7.* Classification accuracy with different values of $\alpha$ and $\rho$

| DATASETS | $\alpha \backslash \rho$ | 0.01 | 0.03 | 0.05 | 1 | 2 |
|---|---|---|---|---|---|---|
| DTD | 0 | 67.98 | 67.24 | 69.24 | 67.11 | 65.13 |
| | 0.02 | 70.11 | 71.70 | 70.34 | 69.26 | 68.41 |
| | 0.08 | 71.02 | **71.73** | 70.88 | 70.01 | 69.20 |
| | 0.5 | 70.43 | 70.86 | 70.61 | 69.23 | 69.01 |
| | 1.5 | 69.67 | 70.93 | 70.93 | 68.61 | 67.14 |
| | 3 | 68.01 | 68.32 | 70.24 | 67.31 | 66.09 |
| DMLAB | 0 | 51.86 | 51.04 | 51.92 | 51.22 | 49.67 |
| | 0.02 | 52.33 | 53.24 | **53.61** | 51.09 | 49.30 |
| | 0.08 | 53.01 | 53.26 | 52.73 | 50.98 | 48.12 |
| | 0.5 | 52.12 | 52.87 | 53.16 | 48.11 | 46.87 |
| | 1.5 | 50.02 | 49.64 | 51.27 | 47.19 | 44.62 |
| | 3 | 48.96 | 48.62 | 48.64 | 46.23 | 44.11 |

dominate the classification loss. However, as shown in Table 6, IBDR maintains robust performance across a reasonable range of $\alpha$.

### A.3. Hyperparameter Sensitivity

IBDR depends on two key hyperparameters: $\alpha$ and $\rho$. In this section, we evaluate IBDR's performance on the DTD and SVHN datasets across various values of $\alpha$ and $\rho$ to assess the algorithm's robustness with respect to these parameters. As discussed in Section A.2, $\alpha$ controls the degree to which we promote diversity among particles. However, larger values of $\alpha$ may cause instability, and we found that $\alpha = 0.02$ works well as a default across all experiments. Conversely, $\rho$ serves as the step size for the ascent step, and setting it too high can also destabilize the model. Nonetheless, as shown in Table 7, our method remains robust over a reasonable range of both hyperparameters, demonstrating a desirable level of stability.

## B. Experimental Details

The experiments were conducted using PyTorch on a Tesla V100 GPU with 40GB of RAM. We set the hyperparameters as follows: $\alpha = 0.02$, $\beta = 10^{-4}$, and $\rho \in \{0.01, 0.03, 0.05\}$, with $\rho$ tuned using the standard validation set. For optimization, we used stochastic gradient descent (SGD) as the base optimizer, combined with a cosine annealing learning rate scheduler, 500 warm-up steps, a momentum of 0.9, and no weight decay.

### B.1. Datasets

B.1.1. IMAGE CLASSIFICATION

The VTAB-1K (Visual Task Adaptation Benchmark) is a diverse and challenging image classification/prediction suite consisting of 19 datasets from various domains. VTAB-1K covers various tasks across different semantics and object categories and is designed to evaluate how well pre-trained models can adapt to various visual tasks by fine-tuning on small datasets. Specifically, in the VTAB-1K setting, only **1,000 training examples** are provided for each task, making it challenging to fine-tune models with limited data.

Some key features of `VTAB-1K` including of:

- **Diverse tasks:** `VTAB-1K` consists of a variety of visual tasks that fall into three broad categories:
    - *Natural:* Tasks derived from natural images, such as classification of real-world objects. This category consists of `CIFAR100`, `Caltech101`, `DTD`, `Oxford Flowers`, `Pets`, `SVHN`, and `Sun397` datasets.
    - *Specialized:* Tasks from specific domains requiring more fine-tuned understanding (e.g., satellite imagery, medical images). This category consists of `Patch Camelyon`, `EuroSAT`, `Resics45`, and `Diabetic Retinopathy` datasets.
    - *Structured*: Tasks involving synthetic or abstract images that require understanding of structured patterns (e.g., depth estimation, object counting). This category consists of `SmallNORB`, `DMLab`, `dSprites`, and `KITTI` datasets.

- **Evaluation Process:** Models are pre-trained on large datasets (e.g., `ImageNet-21K`) and then fine-tuned using only 1,000 examples from each task in `VTAB-1K`, with an official 80/20 train-validation split. The performance is evaluated based on accuracy or other task-specific metrics, testing the model's adaptability and generalization.

### B.1.2. COMMONSENSE REASONING

For more details on the size of the training set and the number of labels for each commonsense reasoning dataset, we refer to Appendix **B.3** of (Wang et al., 2024).

### B.2. Data Augmentations

### B.2.1. IMAGE CLASSIFICATION

Our implementation is based on the repository V-PETL. For each dataset, we use the following data augmentation:

- For `Caltech101, CIFAR100, Clevr-Dist, Dsprites-Loc, Dsprites-Ori, SmallNorb-Azi, SmallNorb-Ele`:

```
self.transform_train = transforms.Compose([
    transforms.Resize((224, 224)),
    transforms.ToTensor(),
    transforms.Normalize(mean=[0.485, 0.456, 0.406],
                         std=[0.229, 0.224, 0.225])])
self.transform_test = transforms.Compose([
    transforms.Resize((224, 224)),
    transforms.ToTensor(),
    transforms.Normalize(mean=[0.485, 0.456, 0.406],
                         std=[0.229, 0.224, 0.225])])
```

- For `Clevr-Count, Diabetic Retinopathy, DMLab, DTD, EuroSAT, KITTI, Flowers102, Pets, Patch Camelyon, Resisc45, Sun397, SVHN`:

```
from timm.data import create_transform
self.transform_train = create_transform(
        input_size=(224, 224),
        is_training=True,
        color_jitter=0.4,
        auto_augment='rand-m9-mstd0.5-inc1',
        re_prob=0.0,
        re_mode='pixel',
        re_count=1,
        interpolation='bicubic',
    )
aug_transform.transforms[0] = transforms.Resize((224, 224),
                                        interpolation=3)
```

```
self.transform_test = transforms.Compose([
        transforms.Resize((224, 224)),
        transforms.ToTensor(),
        transforms.Normalize(mean=[0.485, 0.456, 0.406],
                            std=[0.229, 0.224, 0.225])])
```

### B.2.2. COMMONSENSE REASONING

For more details on the data augmentation used for the commonsense reasoning task, we refer to (Wang et al., 2024).

## C. All Proof

### C.1. Proof of Theorem 4.1:

We restate the theorem

**Theorem C.1.** *Let $\delta \in (0, 1)$. With the probability at least $1 - \delta$ over the choice of $\mathcal{S} \sim \mathcal{D}^N$, we have*

$$\mathcal{L}_{\mathcal{D}}\left(Q^K\right) \leq \min_{\lambda \geq 0}\left\{\lambda\rho + \mathbb{E}_{\boldsymbol{\theta} \sim Q^K}\left[\max_{\boldsymbol{\theta}'}\left\{\mathcal{L}_{\mathcal{S}}(\boldsymbol{\theta}') - \lambda c^K(\boldsymbol{\theta}, \boldsymbol{\theta}')\right\}\right]\right\} + L\sqrt{\frac{KD_{KL}\left(Q\|P\right) + \log\frac{1}{\delta}}{2N}}, \qquad (6)$$

*where $\mathcal{D}$ is the general data/label distribution, $L$ is the upper-bound of the loss function $\ell(\boldsymbol{\theta}; x, y)$, and $c^K(\boldsymbol{\theta}, \boldsymbol{\theta}') = \frac{1}{K}\sum_{i=1}^{K} c(\theta_i, \theta_i')$ represents a distance/divergence between two models.*

*Proof.* Under the assumption that the loss $\ell(\boldsymbol{\theta}; x, y) \leq L$ is bounded by $L > 0$, it follows from **Theorem 4.1** developed by (Alquier et al., 2016) that for any $\beta > 0$

$$\mathcal{L}_{\mathcal{D}}\left(Q^K\right) \leq \mathcal{L}_{\mathcal{S}}\left(Q^K\right) + \frac{1}{\beta}\left[D_{KL}\left(Q^K\|P^K\right) + \log\frac{1}{\delta} + \frac{\beta^2 L^2}{8N}\right]. \qquad (7)$$

Consider the term $D_{KL}(Q^K, P^K)$, which is equal to:

$$\begin{aligned}
D_{KL}(Q^K \| P^K) &= \int_{\theta_1,\ldots,\theta_K} Q^K(\theta_1,\ldots,\theta_K) \log\frac{Q^K(\theta_1,\ldots,\theta_K)}{P^K(\theta_1,\ldots,\theta_K)} d\theta_1 d\theta_2 \cdots d\theta_K \\
&= \int_{\theta_1,\ldots,\theta_K} \prod_{k'=1}^{K} Q(\theta_{k'}) \log\frac{\prod_{k=1}^{K} Q(\theta_k)}{\prod_{k=1}^{K} P(\theta_k)} d\theta_1 d\theta_2 \cdots d\theta_K \\
&= \sum_{k=1}^{K} \int_{\theta_k} Q(\theta_k) \log\frac{Q(\theta_k)}{P(\theta_k)}\left(\prod_{k'\neq k}\int_{\theta_{k'}} Q(\theta_{k'})d\theta_{k'}\right)d\theta_k \\
&= \sum_{k=1}^{K} \int_{\theta_k} Q(\theta_k) \log\frac{Q(\theta_k)}{P(\theta_k)}d\theta_k \\
&= KD_{KL}(Q\|P)
\end{aligned}$$

By choosing $\beta = \sqrt{8N}\frac{\sqrt{D_{KL}(Q^K\|P^K) + \log\frac{1}{\delta}}}{L}$ in Eq. (7), the RHS becomes $\sqrt{\frac{D_{KL}(Q^K\|P^K) + \log\frac{1}{\delta}}{2N}} \times L$. Therefore:

$$\mathcal{L}_{\mathcal{D}}\left(Q^K\right) \le \mathcal{L}_{\mathcal{S}}\left(Q^K\right) + L\sqrt{\frac{D_{\mathrm{KL}}\left(Q^K\|P^K\right) + \log\frac{1}{\delta}}{2N}}$$

$$= \mathcal{L}_{\mathcal{S}}\left(Q^K\right) + L\sqrt{\frac{KD_{\mathrm{KL}}\left(Q\|P\right) + \log\frac{1}{\delta}}{2N}}$$

$$\le \max_{\tilde{Q}^K:W_c\left(\tilde{Q}^K,Q^K\right)\le\rho} \mathcal{L}_{\mathcal{S}}\left(\tilde{Q}^K\right) + L\sqrt{\frac{KD_{\mathrm{KL}}\left(Q\|P\right) + \log\frac{1}{\delta}}{2N}}.$$

According to (Blanchet & Murthy, 2019), the *duality problem* of the DRO problem implies that

$$\max_{\tilde{Q}^K:W_c\left(\tilde{Q}^K,Q^K\right)\le\rho} \mathcal{L}_{\mathcal{S}}\left(\tilde{Q}^K\right) \le \min_{\lambda\ge 0}\left\{\lambda\rho + \mathbb{E}_{\boldsymbol{\theta}\sim Q^K}\left[\max_{\boldsymbol{\theta}'}\left\{\mathcal{L}_{\mathcal{S}}\left(\boldsymbol{\theta}'\right) - \lambda c\left(\boldsymbol{\theta},\boldsymbol{\theta}'\right)\right\}\right]\right\}.$$

Therefore, we conclude that:

$$\mathcal{L}_{\mathcal{D}}\left(Q^K\right) \le \min_{\lambda\ge 0}\left\{\lambda\rho + \mathbb{E}_{\boldsymbol{\theta}\sim Q^K}\left[\max_{\boldsymbol{\theta}'}\left\{\mathcal{L}_{\mathcal{S}}\left(\boldsymbol{\theta}'\right) - \lambda c\left(\boldsymbol{\theta},\boldsymbol{\theta}'\right)\right\}\right]\right\} + L\sqrt{\frac{KD_{\mathrm{KL}}\left(Q\|P\right) + \log\frac{1}{\delta}}{2N}}. \qquad (8)$$

$\square$

## C.2. Proof of Corollary 4.2:

**Corollary C.2.** *Given a metric $d$ over the model space, consider the following cost function $c$*

$$c(\boldsymbol{\theta},\boldsymbol{\theta}') = \begin{cases} d(\boldsymbol{\theta},\boldsymbol{\theta}') & \text{if } d(\boldsymbol{\theta},\boldsymbol{\theta}') \le \rho \\ +\infty & \text{otherwise.} \end{cases}$$

*With the probability at least $1 - \delta$ over the choice of $\mathcal{S} \sim \mathcal{D}^N$, we have*

$$\mathcal{L}_{\mathcal{D}}\left(Q^K\right) \le \mathbb{E}_{\boldsymbol{\theta}\sim Q^K}\left[\max_{\boldsymbol{\theta}'\in\mathcal{B}_\rho(\boldsymbol{\theta})} \mathcal{L}_{\mathcal{S}}\left(\boldsymbol{\theta}'\right)\right] + L\sqrt{\frac{KD_{KL}\left(Q\|P\right) + \log\frac{1}{\delta}}{2N}},$$

*where we define the ball $\mathcal{B}_\rho(\boldsymbol{\theta}) := \{\boldsymbol{\theta}' : d(\boldsymbol{\theta},\boldsymbol{\theta}') \le \rho\}$.*

*Proof.* This result follows directly from the fact that

$$\max_{\boldsymbol{\theta}'}\left\{\mathcal{L}_{\mathcal{S}}\left(\boldsymbol{\theta}'\right) - \lambda c\left(\boldsymbol{\theta},\boldsymbol{\theta}'\right)\right\} = \max_{\boldsymbol{\theta}':d\left(\boldsymbol{\theta},\boldsymbol{\theta}'\right)\le\rho} \mathcal{L}_{\mathcal{S}}\left(\boldsymbol{\theta}'\right)$$

because $c\left(\boldsymbol{\theta},\boldsymbol{\theta}'\right)$ becomes $+\infty$ when $\boldsymbol{\theta}' \notin \mathcal{B}_\rho\left(\boldsymbol{\theta}\right) := \{\boldsymbol{\theta}' : d\left(\boldsymbol{\theta},\boldsymbol{\theta}'\right) \le \rho\}$. Apply this result to Theorem 4.1, the outer minimization in Eq. (8) is obtained when $\lambda = 0$, which concludes the proof. $\square$

## C.3. Proof of Corollary 4.3:

**Corollary C.3.** *With the probability at least $1 - \delta$ over the choice of $\mathcal{S} \sim \mathcal{D}^N$, we have*

$$\mathcal{L}_{\mathcal{D}}\left(Q^K\right) \le \min_{\lambda \ge 0} \left\{ \lambda\rho + \mathbb{E}_{\theta_{1:K} \sim Q} \left[ \max_{\theta'_{1:K}} \left\{ \frac{\sum_{i=1}^K l(\theta'_i; x, y)}{K} + \alpha l_{div}(\theta'_{1:K}; x, y) - \frac{\lambda}{K} \sum_{i=1}^K c(\theta_i, \theta'_i) \right\} \right] \right\}$$

$$+ L\sqrt{\frac{\sum_{i=1}^K \|\mu_i\|^2 + Kd(\sigma - \log\sigma) + 2\log\frac{1}{\delta}}{4N}},$$

where $\mathcal{D}$ is the general data/label distribution, $L$ is the upper-bound of the loss $\ell$, and $d$ is the model size.

*Proof.* This result follows directly from Theorem 4.1 and from the fact that

$$\mathrm{D_{KL}}\left(Q \| P\right) = \mathrm{D_{KL}}\left(\frac{1}{K}\sum_{i=1}^K \mathcal{N}\left(\mu_i, \sigma^2\mathbb{I}\right) \| \mathcal{N}\left(\mathbf{0}, \mathbb{I}\right)\right) \le \frac{1}{K}\sum_{i=1}^K \mathrm{D_{KL}}\left(\mathcal{N}\left(\mu_i, \sigma^2\mathbb{I}\right) \| \mathcal{N}\left(\mathbf{0}, \mathbb{I}\right)\right)$$

$$= \frac{1}{2K}\sum_{i=1}^K \left(\|\mu_i\|^2 + d\sigma - d\log\sigma\right).$$

$\square$

