# OpenReview forum: "Promoting Ensemble Diversity with Interactive Bayesian Distributional Robustness for Fine-tuning Foundation Models"
_ICML.cc/2025/Conference — ICML 2025 poster_

### Official Review · Reviewer_KcS3 · 2025-03-02

**Overall Recommendation:** 4

**Summary:**

The authors propose a new Bayesian inference framework called “Interactive Bayesian Distributional Robustness” (IBDR). IBDR is designed to improve the quality and diversity of model ensembles by modelling interactions between individual models in the ensemble in order to prevent them from collapsing into similar solutions. The proposed IBDR framework uses a dual optimization procedure that promotes both distributional robustness and model diversity. The authors demonstrate empirically that using the IBDR framework leads to improved performance on image classification and common-sense reasoning tasks, where IBDR outperforms baseline methods on the majority of tasks.

## update after rebuttal
As I stated in my comment below, the authors sufficiently answered all of my questions in their rebuttal and provided additional analysis that I think will strengthen the paper. I have increased my score accordingly.

**Claims And Evidence:**

All claims made in the submission are supported by clear convincing evidence. In particular, the large number of tasks and baseline comparisons done in the experiments section provide clear empirical evidence of claims made regarding the performance improvements obtained by using IBDR.

**Essential References Not Discussed:**

There is no essential related work missing from the paper as far as I am aware.

**Experimental Designs Or Analyses:**

I did check the validity of the experimental design. In particular, the authors provided a link to their anonymized codebase which made it easy to check the validity of their experimental setup by looking directly at the code they ran to produce the results provided in the paper. As far as I am aware, the experimental setup is valid.

**Methods And Evaluation Criteria:**

Yes, the authors selected a relevant set of tasks in image classification and common-sense reasoning, and compared IBDR directly to all relevant baselines in the literature that I am aware of.

**Other Comments Or Suggestions:**

1. Typo in running title: “Promoting Ensemble Diversity with Interactive Bayesian Distributional s for Fine-tuning Foundation Models”? What is s?

2. Section title: “B.2. Data augmentations”, augmentations should be capitalized for consistency with other section titles in appendix.

**Other Strengths And Weaknesses:**

Main Strengths:

1. Novelty: IBDR is a novel Bayesian inference framework that introduces a novel approach to deal with the model collapse problem - one of the most relevant problems in this area in the literature.

2. Theoretical motivation: The authors provide strong theoretical evidence to motivate their IBDR framework (i.e. Theorem 4.1).

3. Empirical evidence: The authors provide compelling empirical evidence to demonstrate that IBDR outperforms other state of the art methods in the literature across a large number of benchmark tasks.

Potential Weaknesses:

1. What is the limit on IBDR performance improvement as the number of particles increases?
The authors mention that as the number of particles is increased, the ensemble quality is improved, thus improving performance. However, there is an inherent tradeoff here because increasing the number of particles also increases the runtime linearly. Tables 4 and 5 demonstrate this accuracy-runtime tradeoff with empirical results for 1, 2, 4, and 8 particles. The authors use this result to provide justification of their choice to use 4 particles to balance this tradeoff - which I do agree is a reasonable choice given these results. However, it is clear from Table 4, that up to 8 particles, the performance continues to improve as the number of particles increases. This begs the question: what is the limit on the number of particles after which adding additional particles no longer improves performance? I.e. would accuracy continue to increase as we add up to 20 particles? Up to 100 particles? There must be some limit where adding more particles no longer continues to improve accuracy? I therefore think that it would be very interesting and informative if the authors were to extend the results in Tables 4 and 5 to include larger numbers of particles, ideally up to the number of points where this “limit” on continued accuracy improvement is reached and we see that the accuracy stops improving. This would be useful to demonstrate the best possible performance of IBDR in cases when practitioners are willing to pay higher runtime costs to achieve the best possible accuracy.

2. Runtime of IBDR compared to baselines?
Table 5 provides runtimes per epoch for IBDR with different numbers of particles. However, I think it would also be good to include the runtime per epoch for all baseline methods compared to in Tables 1-3. This would demonstrate the accuracy-runtime tradeoff when choosing between IBDR and these baseline approaches.

**Questions For Authors:**

Please see the two specific questions listed above under “weaknesses”.

**Relation To Broader Scientific Literature:**

The authors show that IBDR improves performance of ensemble learning. This is relevant to the broader community because ensemble learning is very relevant in the literature, especially for applications such as fine-tuning foundation models. Additionally, mode collapse is a very well-studied and relevant problem in the literature. The authors propose a novel and interesting way to address this problem of mode collapse that I think will be of interest to the broader research community. Additionally, the authors demonstrate that IBDR outperforms relevant baselines in the literature on a variety of relevant benchmark tasks, providing motivation for researchers to use IBDR in practice.

**Theoretical Claims:**

The authors provide proofs of Theorem 4.1, Corollary 4.2, and Corollary 4.3 in the appendix (section C). I read these proofs thoroughly. As far as I am aware, the proofs are correct and accurately prove the three relevant claims.

---

> ### Author Rebuttal · Authors · 2025-04-01
>
> We would like to thanks reviewer KcS3 for their supportive review and feedback. We would like to address some question and concern as follow:
> 1. **What is the limit on IBDR performance improvement as the number of particles increases?**
> Thank you for the helpful suggestion. Following your comment, we conducted additional experiments using a greater number of particles. The results are reported in the table below for your convenience.
> As shown, the performance of our method generally improves as the number of particles increases. Notably, we observe a slight performance gain when using 12 particles. However, this appears to be the upper limit of improvement—using 16 particles yields comparable or even slightly lower performance than using 12 or 8 particles.
> | Accuracy         | Camelyon | EuroSAT | Resics45 | Retinopathy |
> |-|-|-|-|-|
> | 1p            | 82.4     | 93.1    | 84.2     | 73.8        |
> | 2p            | 84.8     | 93.9    | 86.6     | 74.3        |
> | 4p            | 85.1     | 95.0    | 87.3     | 76.5        |
> | 8p            | 85.8     | 95.5    | 87.4     | 77.0        |
> | 12p            | 86.1     | 95.3    | 87.6     | 77.3        |
> | 16p            | 85.9     | 95.0   | 87.6     | 77.2        |
> 2. **The accuracy-runtime tradeoff when choosing between IBDR and other baseline approaches.**
>
> We thank the reviewer for the insightful suggestion. We have already conducted experiments comparing the runtime of our method with several baseline approaches. As shown in the table below, our method has a slightly higher runtime than some baselines such as SA-BNN and SVGD. Deep Ensemble, being the simplest among the compared methods, achieves the fastest runtime—this aligns with its relatively lower accuracy, as reported in Table 1 in our paper. These results highlight the performance-runtime trade-off among different methods, as the reviewer suggested. We will extend our runtime analysis to include all datasets and baselines, and incorporate the findings into the revised version.
>
> | Model           | Camelyon | EuroSAT | Resisc | Retinopathy |
> |-----------------|----------|---------|--------|-------------|
> | SA-BNN          | 128      | 121     | 125    | 122         |
> | SVGD            | 124      | 118     | 121    | 120         |
> | Deep Ensemble   | 67       | 64      | 66     | 66          |
> | IBDR (4 particles) | 158   | 149     | 156    | 151         |
>
> 3. **Regarding some typos in our paper**
> - Typo in **running title** and in **Section title: “B.2. Data augmentations”**:
> Thank you for the helpful suggestions. We will correct the typo in the running title by removing the extraneous "s". Additionally, we will capitalize “Augmentations” in Section B.2 to ensure consistency with the formatting of other appendix section titles.

---

> > ### Comment · Reviewer_KcS3 · 2025-04-01
> >
> > Thank you for providing the additional analysis to answer my two questions. I think that adding these two analyses will strengthen the paper. I have updated my score to reflect this.

---

> > > ### Author Response · Authors · 2025-04-02
> > >
> > > We sincerely appreciate your recognition of our work and the time and effort you have dedicated as a reviewer. We will incorporate these analyses into the final version of our paper.

---

### Official Review · Reviewer_yWcp · 2025-03-04

**Overall Recommendation:** 3

**Summary:**

The authors introduce a method to encourage diversity in an ensemble of Bayesian neural-net particles.
To achieve this, they combine results from distributional robustness and determinantal point processes
to derive a PAC-Bayesian-style upper bound on their target objective. An approximation of this bound becomes
 their training objective.
As the title already suggests, the paper focuses solely on finetuning existing models and is evaluated extensively on both image classification and LLM reasoning tasks.

___
_Post-rebuttal update: Switched from weak reject to weak accept given the rebuttal._

**Claims And Evidence:**

The claims are broadly consistent and provided with sufficient evidence. Some restrictions are
- Claims regarding uniqueness. l195 claims that "conventional Bayesian frameworks" can't enforce diversity (see comments on related work below)
- Despite general derivations and theoretical applicability, the method is evaluated only for the subtask of fine-tuning and lacks a proper empirical evaluation of its general statements.  (see below)

**Essential References Not Discussed:**

Although the paper belongs to the field of particle-based Bayesian ensembles, its discussion of the field
is mostly limited to varying MCMC methods and the claim that _"conventional Bayesian frameworks" (l198)_ lack
the required mechanisms.
Other Bayesian approaches, e.g., the literature which can be summarized under the keyword "repulsive ensembles" (D'Angelo & Fortuin, 2021) is completely omitted.


_____
D'Angelo and Fortuin, Repulsive deep ensembles are bayesian, NeurIPS 2021

**Experimental Designs Or Analyses:**

To evaluate fine-tuning results for classification and common-sense reasoning, the experimental design is adequate and extensive with numerous baselines and ablations.

**Methods And Evaluation Criteria:**

Both, the methods and evaluation criteria make sense given the restriction the authors impose on themselves in the title, i.e., fine-tuning.
However, the abstract and most of the theory are written and presented as a general new method for particle-based ensemble learning. The experimental setup is not able to evaluate that.

**Other Comments Or Suggestions:**

- Please follow the ICML style guide when submitting to ICML and use `\citet` and `\citep` whenever appropriate
- l370: _"According to Table 1, IBDR outperforms all baselines by large margins."_ -> That is not what the table shows, there are even situations where IBDR is outperformed by large margins. What is shows is that IBDR outperforms the baselines on average.

### Typos
- l162 right column is missing a minus sign in the exponential
- Running title is broken

**Other Strengths And Weaknesses:**

## Strengths
- clear and understandable writing style


## Weaknesses
- Apart from those already discussed, the major weakness of the paper is its reference section. Many papers point to arxiv preprints instead of the published versions (starting, e.g., with Abbas et al., who published their work in ICML 2022).
Others appear multiple times, e.g., Foret et al., Nguyen et al., Welling et al.,... .
The most drastic case is the Lora paper reference. It is referenced once as "Hu, E. and et al.", then once more with the complete author list "Hu, E. J., Shen..." and finally as Bartlett et al., 2023. The claimed author list is disjoint from the Lora authors, and the arxiv reference finally points to a completely unrelated Reinforcement learning paper. Given that there seems to be a researcher called Lora Bartlett that could mean that an llm was used at least for some of the references.

**Questions For Authors:**

- Why are the baselines between the two experimental setups different?

**Relation To Broader Scientific Literature:**

Except for the crucial omission discussed in the next section the paper's relation to the broader literature
is properly discussed.

**Theoretical Claims:**

All theoretical claims are supported by detailed enough proofs whose correctness I checked.
An exception is the step from theory to practice, which is not justified, i.e., from Corollary 4.3 to equation (5).
While the authors claim this to be _"a minor relaxation"_ (l299) they lack arguments why (5)
is still supposed to be a valid upper bound as it contains a multitude of changes.
Additionally, the theory depends on the fact that the loss function has to be bounded. As far as I can see this constraint is dropped completely during application making leaving the theoretical foundation of the approach on a rather weak foundation.

---

> ### Author Rebuttal · Authors · 2025-04-01
>
> We thank the reviewer for your feedback. We provide detailed responses to the main concerns as follows:
> 1. **l195: 'conventional Bayesian frameworks' can't enforce diversity**
> - Indeed, in Section 3.3, we introduce what we meant by traditional Bayesian framework. Given a training set $S$, we have the closed from for the posterior $p(\theta|\mathcal{S})$. To handle this posterior, variational approach aims to learn an approximate posterior $q(\theta)$ using the ELBO. At the end, we sample $K$ models $\theta_1,\dots,\theta_K$ from $q(\theta)$ for ensemble the predictions. Certainly, this framework does not allow the particle interaction.
> - Our main contribution is the joint distribution $Q^K$, which models interactions among particles and enables distributional robustness. This links to sharpness-aware minimization and allows flexible definitions of interaction (e.g., prediction divergence), unlike prior work like SVGD, wherein particle diversity is primarily enforced through similarity kernels over weight vectors.
>
> 2. **The method is evaluated only for the subtask of fine-tuning ...**
> - A key limitation of ensemble-based methods is the need to store and train multiple models, which becomes computationally expensive when dealing with large-scale models.
> Besides, in the era of model finetuning, where fine-tuning pre-trained models is a dominant paradigm, our method becomes particularly advantageous. It mitigates the primary drawbacks of traditional ensemble approaches, by requiring only the storage of multiple lightweight adapters rather than full models.
> - Following your suggestions, we have also conducted additional experiments involving training from scratch on CIFAR100 as follows:
> |Accuracy|Resnet18|Resnet34|
> |-|-|-|
> |Standard Training|76.29|77.31|
> |IBDR (4 particles)|77.45|77.92|
> 3. **... why (5) is still supposed to be a valid upper bound as it contains a multitude of changes**
> - Eq (5) is a relaxation of the one in Corolarry 4.3 that keeps its spirit. Particularly, for the one in Corolarry 4.3, given $\theta_{1:K}\overset{iid}{\sim}Q$, we need to find out the perturbation $\theta_{1:K}^{'}$ all in once that maximizes the loss while staying closely to $\theta_{1:K}$ via minimizing $\frac{\lambda}{K}\sum_{k=1}^{K}c(\theta_{k},\theta_{k}^{'})$. In the relaxation version, we isolate $\theta_k$ and $\theta_k^{'}$ in the sense that we find the perturbation $\theta_{k}^{'}$ around $\theta_k$ by maximizing the loss and minimizing the distance $c(\theta_k^{'}, \theta_k)$, while replacing $\theta_{-i}^{'}$ by the fixed $\theta_{-i}$. This is tolerable because $\theta_{-i}^{'}$ stays closely to $\theta_{-i}$. Besides being easier to implement, this relaxation still preserves the spirit of the original one.
> 4. **The theory depends on the fact that the loss function has to be bounded...but this constraint is dropped completely during application...**
> - We do not break or violate the bounded loss constraint in our application. In particular, we mainly focus on classification tasks with the CE loss, whose formulation can be simplified as $-\log p_y$ where $y$ is the ground-truth label of $x$ and $p_y$ is the model prediction. We acknowledge that $-\log p_y$ tends to infinity if and only if $p_y$ tends to 0. This is impossible due to 2 reasons:
>     - At the beginning of training, the model lacks prior knowledge and typically makes near-uniform predictions. This causes the predicted probability $p$ to fluctuate around $1/M$, where $M$ is the number of classes, rather than approaching zero.
>     - During training, the objective is to increase the predicted probability for the correct class (maximize $p_y$). Therefore, the training process inherently pushes $p_y$ closer to 1, not 0.
> - In addition, we notice that in the SAM paper, the McAllester PAC-Bayes bound was used to develop its theory, which is only applicable to the 0-1 loss. In this paper, we leverage with more advanced PAC-Bayes theorem, leading to a less restrictive family of loss functions.
> 5. **.., paper's discussion of the field is mostly limited to varying MCMC methods. Other Bayesian approaches,... is completely omitted**
> - We will definitely discuss [D'Angelo & Fortuin, 2021] in the revised version. However, we notice that our approach is fundamentally different from these Stein approaches.
> 6. **The LoRA paper reference:**
> - Thanks for your comment. This is indeed an error in our bibtex. We will definitely fix them in the revised version.
> 7. **l370: "Table 1, IBDR outperforms all baselines by large margins..."**
> - We will change the wording. Indeed, Table 1 includes a column labeled AVG, and our method outperforms all other baselines by more than 2% in terms of average accuracy.
> 8. **Why are the baselines between the two experimental setups different**
> - We chose baselines closely related to our method for image classification, and followed BLoB’s setup for commonsense reasoning, including comparisons with popular Bayesian methods in LLMs.

---

> > ### Comment · Reviewer_yWcp · 2025-04-02
> >
> > Thank you for your rebuttal and the answers. The following are only clarifying comments from my side and do not require an answer from the reviewers (unless, of course, they want to).
> >
> > - **On fine-tuning**: I agree that fine-tuning is becoming an important task, yet disagree that it is _"a dominant paradigm"_, the field is still larger than foundational models. In my opinion, ensembles face two problems. (i) storage and computational resources, which you state, but also (ii) the potential of collapsing unto a subset of modes, i.e., losing their diversity.
> > After reading the current work, the reader is left wondering, whether they can use it also in their shallower, more classical, approaches. Your preliminary CIFAR100 results suggest that that is the case, further strengthening the paper.
> >
> > - **valid upper bound**: A _relaxation_ would be to switch to a looser bound that might have some benefits, such as easier training, better numerical stability, etc. If I read the paper and your explanation correctly, what you are doing is an _approximation_ to the true bound. This is completely fine, but should be presented that way.
> >
> > - **_"However, we notice that our approach is fundamentally different from these Stein approaches."_**
> > The point I tried to make was that reading the variational inference paragraph (starting l064) the existence of particle-based variational inference approaches, be they stein-based, repulsive BNNs, etc. is completely omitted.
> >
> > One remaining weakness of the work is that the method tends to struggle with calibration in certain regards. I encourage the authors to explore further improvements in that direction in future work.
> >
> >
> > Given your overall rebuttal and the other reviews, I adjust my score accordingly.

---

> > > ### Author Response · Authors · 2025-04-04
> > >
> > > We appreciate the reviewer’s insightful feedback and valuable suggestions, which will undoubtedly enhance our paper. We will certainly incorporate them into the revised version.

---

### Official Review · Reviewer_NLbp · 2025-03-06

**Overall Recommendation:** 3

**Summary:**

The paper introduces Interactive Bayesian Distributional Robustness, a novel Bayesian inference framework designed to improve ensemble diversity and robustness in fine-tuning foundation models. The core idea of IBDR is to explicitly model interactions between multiple sampled particles in the Bayesian inference process. Unlike traditional Bayesian methods, which treat sampled models as independent, IBDR leverages a joint distribution and introduces a divergence loss to enforce diversity among the sampled models.

**Claims And Evidence:**

1. IBDR enhances ensemble diversity compared to existing Bayesian methods  -- Supported by experiments.
2. IBDR improves robustness through Wasserstein-based distributional optimization  -- Supported by theoretical derivations and empirical results
3. IBDR generalizes across tasks, from vision (ViT) to language models (LLaMA-2) -- Supported by experiments

**Essential References Not Discussed:**

NA

**Experimental Designs Or Analyses:**

The experimental design is well-structured and follows standard machine learning evaluation practices.

**Methods And Evaluation Criteria:**

The accuracy and Expected Calibration Error (ECE) metrics are appropriate for evaluating both predictive performance and model uncertainty.

**Other Comments Or Suggestions:**

See above.

**Other Strengths And Weaknesses:**

While IBDR is evaluated on multiple datasets, computational overhead is not thoroughly analyzed. Given that the method involves interactive Bayesian sampling, a study on efficiency would be valuable.

**Questions For Authors:**

NAAN

**Relation To Broader Scientific Literature:**

- Bayesian Neural Networks: Extends variational inference and Bayesian optimization techniques.
- Sharpness-Aware Minimization: Connects to robustness-aware training methods.
- Distributional Robustness Optimization: Uses Wasserstein-based robustness techniques.

**Theoretical Claims:**

The paper presents several theoretical claims, particularly Theorem 4.1 and Corollary 4.2, which establish an upper bound for the population loss under the IBDR framework. These claims appear mathematically sound and extend prior results in distributional robustness.

---

> ### Author Rebuttal · Authors · 2025-04-01
>
> We appreciate the reviewer’s comments and respond to the key concerns as follows:
>
> **"While IBDR is evaluated on multiple datasets, computational overhead is not thoroughly analyzed. Given that the method involves interactive Bayesian sampling, a study on efficiency would be valuable."**
>
> Thank you for the insightful suggestion. We would like to point out that an analysis of performance and runtime with varying numbers of particles is already provided in Appendix A.1. For your convenience, we report these corresponding tables below.
>
> | Accuracy         | Camelyon | EuroSAT | Resics45 | Retinopathy |
> |---------------|----------|---------|----------|-------------|
> | 1p            | 82.4     | 93.1    | 84.2     | 73.8        |
> | 2p            | 84.8     | 93.9    | 86.6     | 74.3        |
> | 4p            | 85.1     | 95.0    | 87.3     | 76.5        |
> | 8p            | 85.8     | 95.5    | 87.4     | 77.0        |
>
>
> | Runtime (sec/epoch) | Camelyon     | EuroSAT     | Resics45    | Retinopathy  |
> |---------------------|--------------|-------------|-------------|--------------|
> | 1p                  | 51 ± 1.8     | 50 ± 1.5     | 48 ± 1.7     | 51 ± 0.7      |
> | 2p                  | 80 ± 2.1     | 83 ± 2.4     | 93 ± 2.1     | 85 ± 0.9      |
> | 4p                  | 158 ± 4.3    | 161 ± 4.9    | 156 ± 4.1    | 151 ± 2.1     |
> | 8p                  | 220 ± 6.2    | 230 ± 5.7    | 218 ± 7.3    | 246 ± 6.8     |

---

### Official Review · Reviewer_djew · 2025-03-14

**Overall Recommendation:** 3

**Summary:**

This paper introduces a distributionally robust method for Bayesian estimation, aimed primarily at fine-tuning foundation models. Central to the contribution is a term to promote particle diversity during optimization. Theoretical results of the proposed method are provided, and extensive fine-tuning experiments for vision transformers and language models are presented.

## Update after rebuttal

The authors have committed to fixing the errors in the proof of Theorem 4.1 and clarified its claims. This was my primary concern in the submitted manuscript, and thus, I have raised my score from a 1 to a 3 post-rebuttal.

**Claims And Evidence:**

The empirical claims about the performance of IBDR seem well-evidenced. There are several claims about variational inference that I do not agree with (see the "essential references" below), but this is more minor. I also have significant comments about the theory (see below).

**Essential References Not Discussed:**

Several times it is claimed that variational inference methods lack a mechanism to promote diversity. However, approaches like nonlinear Stein VI [1] have very similar explicit forms, optimizing an objective that combines a loss function with a diversity penalty. One may also use mixture variational posteriors, which have recently been shown quite successful in deep learning tasks [2].

**Experimental Designs Or Analyses:**

The experimental design and analysis is appropriate as far as I can tell. As mentioned above, the methods and evaluation criteria are appropriate.

**Methods And Evaluation Criteria:**

The proposed methods and evaluation criteria make sense. Standard fine-tuning tasks are chosen for reasonable foundation models, with strong baselines.

**Other Comments Or Suggestions:**

Some typographical comments:
- (Line 16, Right Column) These citations should use `\citet{}`.
- (Line 150, Right Column) There is a missing space in "cross entropy (CE)".
- (Line 249, Right Column) Strictly speaking, $f(x; \theta_i)$ is generally not a "predictive probability."
- (Line 321, Left Column) "Gradient descend" should be "gradient descent".
- (Appendix C) The notations $D_\text{KL}(Q \parallel P)$ and $D_\text{KL}(Q , P)$ are inconsistently used.

I think it would be interesting to compare results to LoRA with IVON [3], though this was only made publicly available in December and thus falls under "concurrent work."

**Other Strengths And Weaknesses:**

I think clarity can be improved, particularly in the theoretical section.

**Questions For Authors:**

1. What is the meaning of "non-maximal prediction probabilities"?
2. The nonlinear SVGD paper [2] found that the repelling regularizer could be quite important, and in particular settles on an entropy-based regularizer rather than a log-determinant-based one. Have the authors considered other similar choices?

## References
[1] Wang, D., & Liu, Q. (2019). Nonlinear Stein variational gradient descent for learning diversified mixture models. In International Conference on Machine Learning (pp. 6576-6585). PMLR.

[2] Shen, Y., Daheim, N., Cong, B., Nickl, P., Marconi, G. M., Bazan, C., ... & Möllenhoff, T. (2024). Variational learning is effective for large deep networks. arXiv preprint arXiv:2402.17641.

[3] Cong, B., Daheim, N., Shen, Y., Cremers, D., Yokota, R., Khan, M. E., & Möllenhoff, T. (2024). Variational Low-Rank Adaptation Using IVON. arXiv preprint arXiv:2411.04421.

**Relation To Broader Scientific Literature:**

The proposed method relates to the current and relevant topic of fine-tuning foundation models, through an interesting lens of distributional robustness.

**Theoretical Claims:**

I am not an expert in distributionally robust optimization in particular, but was not able to follow all technical claims, and believe presentation can be improved. For example, the proof of Theorem 4.1 states "it follows from the PAC-Bayes theorem developed by (Alquier et al., 2016) [...]" -- but (Alquier et al., 2016) is a 40 page paper with several theorems. It appears that the authors are referring to Theorem 4.1, but then there is a missing logarithm for $1 / \delta$ (which is introduced later). The proof subsequently states "by choosing $\beta = $[...] in Eq. (6)", but there is no $\beta$ in Eq. (6). Assuming this is actually referring to the unlabeled equation above, the stated value of $\beta$ is not correct; unless I'm missing something, even after correcting $1 / \delta \rightarrow \log(1/\delta)$, the correct value of $\beta$ should actually be
$$\beta = \frac{\sqrt{8N}}{L} \sqrt{D_{\text{KL}}(Q^K \parallel P^K) + \log(1 / \delta)}$$
for the algebra to work out. I did not attempt to verify further claims.

---

> ### Author Rebuttal · Authors · 2025-04-01
>
> We thank the reviewer for the constructive feedback and would like to address the concerns as follows:
> 1. **Regarding the Theoretical Claims:**
> - **Regarding the citation of prior work in the proof of Theorem 4.1** : Thank you for the suggestion. Our proof relies on Theorem 2.1 from the cited study. We will clarify this in the revision to better guide readers.
> - **Regarding the $\beta$ value in our proof:** We appreciate the reviewer for poiting out our typos. In line 858, ${1}/{\delta}$ should be $\log\frac{1}{\delta}$. In line 878, the correct value $\beta$ should include a nested square root over the divergence and $\log\frac{1}{\delta}$, as reviewer suggested. Also, “equation (6)” in line 878 should refer to line 858, which is equation (7). We carefully double-check our proof and except these typo, our proof remains true. Specifically, the RHS of this equation (7) actually becomes $\sqrt{\frac{\text{D}_\text{KL}\left(Q^{K} \| P^{K}\right)+\log\frac{1}{\delta}}{2N}}\times L$ with the correct value of $\beta$ above.
> 2. **Regarding our claim about promoting diversity**
> - **Conventional Bayesian frameworks lack explicit mechanisms to model interactions between particles θ1:K during training:** Indeed, in Section 3.3, we introduce what we meant by traditional Bayesian framework. Given a training set $S$, we have the closed from for the posterior $p(\theta|\mathcal{S})$. To handle this posterior, variational approach aims to learn an approximate posterior $q(\theta)$ using the ELBO. At the end, we sample $K$ models $\theta_1,\dots,\theta_K$ from $q(\theta)$ for ensemble the predictions. Certainly, this framework does not allow the particle interaction.
> - **The novelty of our promoting diversity mechanism:**
> Our main contribution is the joint distribution $Q^K$, which models interactions among particles and enables distributional robustness. This links to sharpness-aware minimization and allows flexible definitions of interaction (e.g., prediction divergence), unlike prior work like SVGD, wherein particle diversity is primarily enforced through similarity kernels over weight vectors.
> 3. **Comparison with nonlinear Stein VI approach**:
> - Thanks for pointing out us the nonlinear Stein paper. This is an interesting paper that extends Stein Variational Gradient Descend (SVGD). These two share the same form of the objective function  $\max_{\rho}{ \left(F(\rho)+H(\rho)\right)}$ where $H\left(\rho\right)=-\int\log\rho d\rho$ is the entropy. As pointed out in Eq.(8) in Theorem 1 of the non-linear Stein paper, the second term in this equation relevant to $\nabla_{\theta}k\left(\theta,.\right)$ is known as the repulsive term derived from maximizing the entropy $H\left(\rho\right)$. This term encourages the particles to spread out for avoiding the mode collapse.
> - Differently, our **proposed approach** enables the model interaction through  $l_{div}\left(\theta_{1:K},x,y\right)$. Interestingly, we can define this term appropriately to encourage various kinds of diversity (e.g., diversity in the model span or diversity in model predictions). Evidently, in our practical method, motivated by the theory of Determinantal Point Processes, we propose a diversity loss to encourage the diversity in model predictions that is proven to improve the ensemble performance.
> - Finally, we will cite and discuss the non-linear Stein paper.
>
> 4. **Typographical comments**: We thank the reviewer for catching these typos. We will correct these typos in the revised manuscript.
> 5. **Comparison results with IVON LoRA:** As reported in **Section 5.2** of our paper, we conduct experiments on CommonSense Reasoning, which are also used in IVON-LoRA’s study. For the reviewer’s convenience, we will present our results alongside theirs as follows:
>
> | Accuracy| WG-S | ARC-C | ARC-E | WG-M | OBQA | BoolQ |
> |-------------------|------|-------|-------|------|------|--------|
> | IBDR (ours)|  72.51|   70.56|  86.95 |   76.46 |   84.60 |   86.89     |
> | IVON-LoRA |  72.1|  69.9|  87.5     |  76.6    |  80.9    | 86.1       |
>
> | ECE  | WG-S | ARC-C | ARC-E | WG-M | OBQA | BoolQ |
> |-------------------|------|-------|-------|------|------|--------|
> | IBDR (ours)|   24.17   |   21.20    |  9.71|   11.19|  5.82    |1.54 |
> | IVON-LoRA |  27.5    |  25.8|  10.1 |  23.0|   11.2|   5.6|
> 6. **Other Reviewer's Question**
> - **The meaning of "non-maximal prediction probabilities"?**: These are the predicted class probabilities excluding the one assigned to the ground-truth class. For example, if the true class is $y_3$ and the original model prediction probability is $[p_1, p_2, p_3, p_4]$ for classes classes $y_1, y_2, y_3, y_4$, then the non-maximal prediction is $[p_1, p_2, p_4]$
> - **Regarding question about entropy-based regularizer**: In our proposed approach, we use the diversity loss $l_{div}\left(\theta_{1:K},x,y\right)$ to encourage the diversity of model predictions instead of the model weight diversity from the entropy maximization.

---

> > ### Comment · Reviewer_djew · 2025-04-02
> >
> > Thanks for the reply -- especially the commitment to fixing the errors in the theoretical results, which were my primary concern. I've raised my score accordingly.

---

> > > ### Author Response · Authors · 2025-04-03
> > >
> > > Thank you for recognizing the contributions of our work and for your constructive feedback. We will carefully incorporate your suggestions to strengthen the final revision.

---

### Decision · Program_Chairs · 2025-05-01

**Decision:**

Accept (poster)

**Comment:**

This paper proposes a novel Bayesian inference framework called Interactive Bayesian Distributional Robustness (IBDR) for fine-tuning foundation models, which aims to improve ensemble diversity and robustness by modeling interactions between multiple sampled particles. The reviewers praised the paper for its novelty, theoretical motivation, and compelling empirical evidence. The main criticisms were that the method might struggle with calibration, the limit of IBDR performance improvement as the number of particles increases was not fully analyzed, and some reviewers found the theoretical results and relation to other Bayesian approaches to be unclear.

The authors responded by addressing these concerns and providing additional analyses and explanations. The reviewers indicated that the authors' rebuttal has addressed most of the criticisms, particularly regarding the analysis of the limit of IBDR performance improvement and the runtime comparison with baseline methods. Therefore, the reviewers unanimously agreed to accept the paper. We would still recommend that the authors take the reviewers' feedback into account when preparing the camera-ready version.